# The human VGLUT3-pT8I mutation elicits uneven striatal DA signaling, food or drug maladaptive consumption in male mice

Mathieu Favier [1,19] ✉, Elena Martin Garcia [2,3,4,19], Romain Icick[5,6,7,8,19], Camille de Almeida[9,19], Joachim Jehl [9,10,19], Mazarine Desplanque[9], Johannes Zimmermann[11], Annabelle Henrion[12,13], Nina Mansouri-Guilani[9], Coline Mounier[1], Svethna Ribeiro[1], Fiona Henderson [9], Andrea Geoffroy[9], Sebastien Mella[9], Odile Poirel[9], Véronique Bernard [9], Véronique Fabre[9], Yulong Li [14], Christian Rosenmund [11], Stéphane Jamain [12,13], Florence Vorspan[5,6,7], Alexandre Mourot [9,10], Philibert Duriez[15], Leora Pinhas[16], Rafael Maldonado [2,3,4,19], Nicolas Pietrancosta[9,17,18,19], Stéphane Daumas[9,19] & Salah El Mestikawy [1,9,19] ✉

Cholinergic striatal interneurons (ChIs) express the vesicular glutamate transporter 3 (VGLUT3) which allows them to regulate the striatal network with glutamate and acetylcholine (ACh). In addition, VGLUT3-dependent glutamate increases ACh vesicular stores through vesicular synergy. A missense polymorphism, VGLUT3-p.T8I, was identified in patients with substance use disorders (SUDs) and eating disorders (EDs). A mouse line was generated to understand the neurochemical and behavioral impact of the p.T8I variant. In VGLUT3$^{T8I/T8I}$ male mice, glutamate signaling was unchanged but vesicular synergy and ACh release were blunted. Mutant male mice exhibited a reduced DA release in the dorsomedial striatum but not in the dorsolateral striatum, facilitating habit formation and exacerbating maladaptive use of drug or food. Increasing ACh tone with donepezil reversed the self-starvation phenotype observed in VGLUT3$^{T8I/T8I}$ male mice. Our study suggests that unbalanced dopaminergic transmission in the dorsal striatum could be a common mechanism between SUDs and EDs.

Persistent habitual behaviors and compulsion are considered as common features of substance use disorders (SUDs) and eating disorders (EDs)[1–5]. The dorsal striatum is central for the transition from reward-guided and goal-directed behaviors to automatic habitual behaviors, and finally to compulsion[6]. Several reports highlight the pivotal and complex role played by cholinergic striatal interneurons (ChIs) in normal and pathological states[7–9]. ChIs are sparsely distributed in the striatum where they represent 1–2% of the neuronal population but account for a dense plexus of varicosities[10]. ChIs express both the vesicular acetylcholine transporter (VAChT) and the

atypical vesicular glutamate transporter type 3 (VGLUT3)[11]. Consequently, ChIs regulate the striatal network with both acetylcholine (ACh) and glutamate (for review[12,13]). Moreover, VGLUT3-dependent glutamate increases the vesicular accumulation of ACh and thus cholinergic tone in the striatum[11].

In mice, selective ablation of VAChT in ChIs (silencing ACh release from ChIs) was initially shown to have only a minor impact on psychostimulant-induced locomotor activity[14]. More recently, VAChT deletion was found to induce facilitation of habit formation and vulnerability to maladaptive eating[15]. In contrast to VAChTcKO mice,

ablation of VGLUT3 results in a marked increase of anxiety phenotype and sensitivity to cocaine reinforcing properties[16,17]. Several studies established that ACh released from ChIs stimulates dopamine (DA) efflux through interactions with nicotinic receptors located on DA fibers and muscarinic receptors[15,17–20]. Conversely, glutamate released by ChIs binds to metabotropic glutamate receptors and inhibits DA release[15,17]. Importantly, this dual and opposite regulation of DA release by ACh/glutamate co-transmission is observed only in the nucleus accumbens and in the dorsomedial striatum (DMS, caudate in humans), but not in the dorsolateral striatum (DLS, putamen in humans)[15]. Finally, Sakae et al. reported an increased frequency of rare variants of VGLUT3 in a cohort of patients with SUDs[17]. One of these variants, the heterozygous *SLC17A8* p.T8I missense mutation, was found in 6 patients with SUDs[17] and in one patient with cocaine addiction and bulimia nervosa (present study). Taken together, all these data suggest a pivotal and complex involvement of ChIs and ACh/glutamate co-transmission in habit formation, SUDs and EDs.

In the present study, the genetic screening for the *SCL17A8* (coding for VGLUT3) mutation was replicated in new patient samples. In parallel, a knock-in mouse line (VGLUT3$^{T8I/T8I}$) was developed to investigate molecular, cellular and behavioral consequences of the VGLUT3-p.T8I variant in male mice. We found that the p.T8I variation did not alter the 3D structure, the amount or distribution of VGLUT3, nor glutamate vesicular accumulation or glutamate release. However, this mutation blunted vesicular synergy, decreasing striatal ACh release on the one hand, and DA release in the DMS but not in the DLS on the other. At the behavioral level, this uneven DA transmission facilitated habit formation, increased cocaine-seeking relapse, and promoted maladaptive eating in binge-like sucrose overconsumption and activity-based anorexia (ABA) models in male mice. Importantly, donepezil (an acetylcholinesterase inhibitor) was effective to reverse the increased self-starvation phenotype observed in VGLUT3$^{T8I/T8I}$ male mice.

These findings suggest that the p.T8I mutation may represent a vulnerability factor in both SUDs and EDs. Furthermore, the present work identifies a mechanism and a potential treatment to alleviate these severe psychiatric disorders.

## Results

### Genetic and clinical characterization of VGLUT3-p.T8I

#### Screening for mutations of the SCL17A8 gene was replicated in new patient samples

**Frequent VGLUT3 variants.** One hundred thirty frequent synonymous/non-coding SNPs (alternate allele frequency >0.01) were analyzed from VGLUT3 gene (see Supplementary dataset for a complete list). Two of them (rs10778052-C and rs11110353-A) showed a significant association with physical symptoms before cocaine use (beta = 0.64 and beta = 1, $p_{uncorrected}$ = 0.000185 and $p_{uncorrected}$ = 0.000374, $p_{corrected}$ = 0.03123 and $p_{corrected}$ = 0.0494, respectively). According to online databases for functional impact assessment (see Supplementary Material), these intronic SNPs were predicted to have little functional impact on gene expression or methylation, chromatin conformation and overall deleteriousness.

**Comparison of p.T8I allelic frequency in patients vs. control samples according to ancestry.** The p.T8I rare variant of VGLUT3 (rs45610843) was identified in patients with SUDs[17]. In the extended clinical sample (EDs + SUDs samples), heterozygous *SLC17A8* p.T8I missense mutation was found in 9/793 cases (1.1%), which were all heterozygous. The genetically-informed sex distribution in the SUDs sample, was 116 (22%) women and 408 (78%) men in total genotyped sample (Supplemental Table S7). There were two (25%) women and six (75%) men in patients who carried p.T8I mutation. The sex distribution between the total sample and p.T8I carriers showed a total absence of sex imbalance associated with p.T8I (Fisher exact test, $p$ = 1). The p.T8I allelic frequency significantly differed between patients and the

population referred in gnomAD (see Supplementary Methods). Patients carried nine risk alleles and 1595 reference alleles (minor allelic frequency (MAF 0.8%) (Fisher's exact test $p = 6.12 \times 10^{-5}$ when compared to the whole gnomAD population). Interestingly, one p.T8I carrier from the EDs sample, a woman with African descent, suffered from both bulimia nervosa and severe cocaine use disorder.

**Clinical characterization of p.T8I variant carriers and of other SLC17A8 (VGLUT3 gene) missense mutations.** In the 363 cocaine use disorder patients (CUD) sample with extended phenotypic data, *SLC17A8* mutations were associated with different total scores on the scale for the assessment of psychotic symptoms (SAPS) and on the delusion subscale (Fig. 1a, Kruskal-Wallis chi-squared = 8.16, df = 2, $p$ = 0.0169, effect size = 0.00897 (small)). In the SUDs patients sample, *SLC17A8* mutations were also associated with reduced prevalence of alcohol use disorders (Fig. 1c, Fisher exact test, $p$ = 0.03). Interestingly, we observed a significant trend toward increasing prevalence of alcohol use disorder ranging from carrier of "other" VGLUT3 mutations (16.7%) to p.T8I carriers (40%) and to VGLUT3 non-mutated patients (62.4%) (trend test, $p$ = 0.013). Figure 1c shows the precise distribution of SAPS total scores as a function of the presence or the absence of p.T8I and/or other *SLC17A8* mutations, confirming that p.T8I carriers had higher scores than both patients without any *SLC17A8* mutation and those with *SLC17A8* mutations other than p.T8I (Wilcoxon tests, p = 0.01787 T8I vs. no SLC17A8 variant and p = 0.04768 T8I vs. other SLC17A8 variants, respectively). To compare variations of VGLUT3 with variations of VGLUT1 and VGLUT2, the 2 genes *SLC17A7* (VGLUT1) and *SLC17A6* (VGLUT2) were also analyzed. As shown in Fig. 1d, missense mutations from *SLC17A7* and *SLC17A6* genes did not significantly alter the behavioral responses to cocaine according to the SAPS scale (Wilcoxon test, $p$ = 0.3126). Previous studies reported that higher doses of cocaine and, especially, more rapid routes of administration (i.e. intravenous or smoking vs snorting) were associated with total SAPS scores[21]. In our sample, p.T8I was not associated with either of these fine-grained clinical variables, which were thus unlikely to bias the association between SAPS and p.T8I (Fisher's Exact Test $p$-value = 1, Supplementary Table 10 and Fig. S1). Two sets of additional analyses were performed. We showed similar group differences using raw (i.e. non-imputed) SAPS scores (Kruskal-Wallis chi-squared = 6.88, df = 2, $p$-value = 0.032, effect size= 0.0094 (Supplementary Table 1 and Fig. S1) as compared to main results. Second, we computed Bayesian factors to seek support for genotype effect on SAPS scores (total and delusions), yielded $BF_{01}$ = 4 for the comparison between p.T8I carriers and patients without *SLC17A8* mutations. This implies that our hypothesis of actually higher SAPS scores in p.T8I vs non *SLC17A8* mutations carriers is four times more probable. These findings provide substantial support to higher SAPS total and delusion scores in p.T8I carriers ($BF_{01}$ can only be computed for pairwise differences, see Supplementary Table 1, Supplementary Fig. 1b).

Although obtained in a small number of carriers, these clinical observations suggest that the p.T8I variant is relatively frequent in patients with African origin and is associated with more severe addictive symptoms related to the response to cocaine. Furthermore, one patient from the EDs sample suffered from both bulimia nervosa and drug abuse (cocaine use disorder). These observations suggest that the p.T8I variant may be associated with both SUDs and EDs.

### The p.T8I mutation does not alter the expression or the targeting of VGLUT3

To explore whether (or not) the p.T8I variant is causally involved in EDs and SUDs, we explored its molecular implications. To determine whether p.T8I alters the expression of VGLUT3, the wildtype (WT) and mutated isoforms were expressed in primary cultures of hippocampal neurons (Fig. 2). As previously reported for the WT isoform[22,23], the VGLUT3-p.T8I isoform was present in the soma, proximal dendrites, in

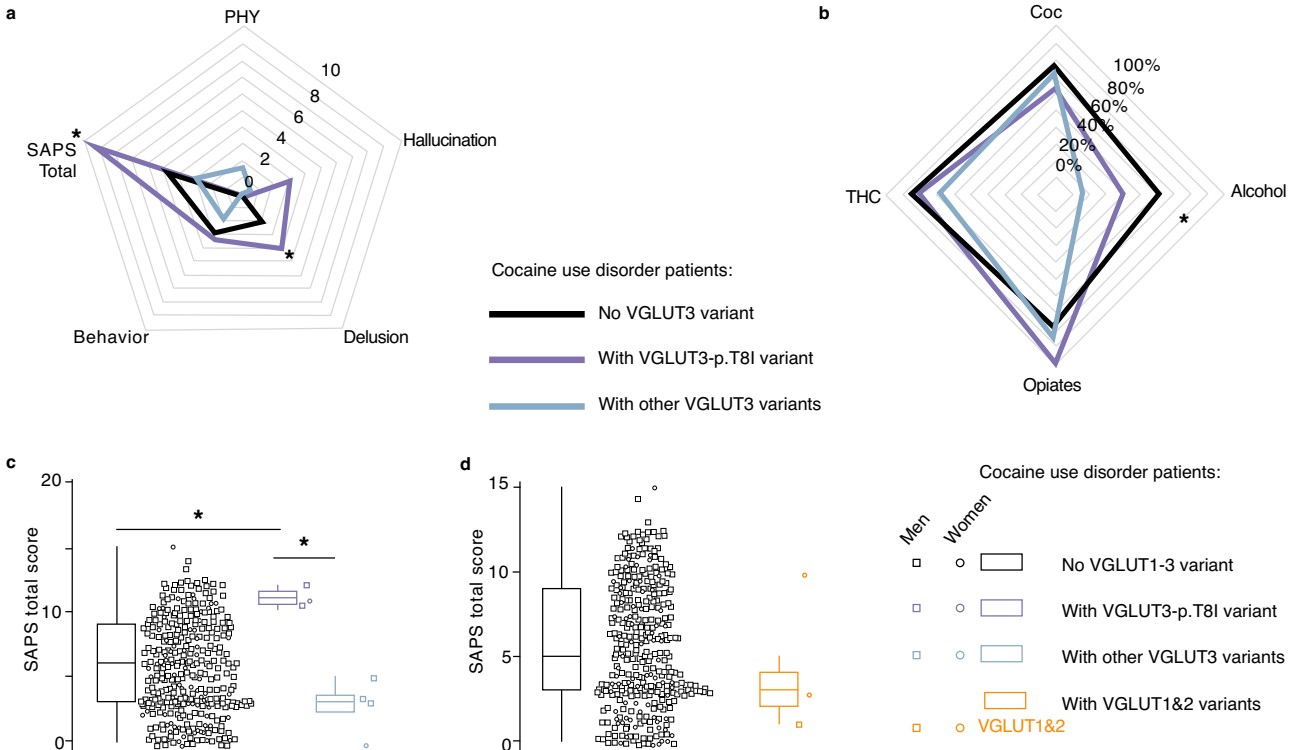

**Fig. 1 | Patients carrying the VGLUT3-p.T8I variant show increased clinical severity of cocaine use disorders.** Scores at the scale for the assessment of psychotic symptoms (SAPS). Total score is the sum of the delusion, hallucination, physical symptoms during cocaine craving (PHY) and behavioral subscales. Uncorrected p-values are shown. VGLUT1 gene, *SLC17A7*; VGLUT2 gene, *SLC17A6*; VGLUT3 gene, *SLC17A8*. **a**, SAPS scores as a function of p.T8I vs other VGLUT3 variant vs. no VGLUT3 mutation in patients with cocaine use disorder (Kruskal-Wallis chi-squared=8.16, df=2, two-sided, $p = 0.0169$ for total score and Kruskal-Wallis chi-squared=8.16, df=2, $p = 0.035$ for delusion subscale). N = 338 biologically independent samples. **b**, Frequency of substance use disorders (SUDs) (Coc, CUD; alcohol; opioids; cannabis, THC) as a function of p.T8I vs other VGLUT3 mutations vs no VGLUT3 mutation in the whole SUDs sample. The presence of p.T8I was not associated with increased prevalence of CUD but with decreased prevalence of alcohol use disorders (two-sided Fisher exact test, $p = 0.03034$). N = 338 biologically independent samples. c, SAPS total score as a function of *VGLUT3* variants

(none *vs*. p.T8I *vs*. others) in patients with CUD score significantly differed across the three genotypes groups (Kruskal-Wallis $p = 0.0169$). p.T8I carriers had significantly higher SAPS score compared to patients without VGLUT3 mutation (two-sided Wilcoxon test, $p = 0.018$), to those with other VGLUT3 mutations (two-sided Wilcoxon test, $p = 0.018$) and to those without any VGLUT mutation (two-sided Wilcoxon test, $p = 0.048$). N = 363 biologically independent samples. **d**, Total score at the SAPS as a function of VGLUT1 or VGLUT2 vs no VGLUT gene variants in CUD patients (two-sided Wilcoxon test =210, p = 0.92). N = 363 biologically independent samples. Source data are provided as a Source Data file. Boxplots are defined as follows (R program defaults), where IQR stands for interquartile range [Q3 (75th percentile value) - Q1 (25th percentile value)]: lower whisker = max(min(x), Q1 − 1.5 * IQR), lower bound of box =25th percentile, center of box =median, upper bound of box =75th percentile, upper whisker =min(max(x), Q3 + 1.5 * IQR). Lack of association with SAPS - delusion not shown for VGLUT3 and VGLUT1-2 variants (see Supplementary Fig. 1b).

axonal processes and varicosities of transfected neurons (Fig. 2a, b). In varicosities, VGLUT3 and VGLUT3-p.T8I colocalized with bassoon, a marker of the active zone (Fig. 2c, d). These experiments suggest that the p.T8I mutation does not alter the targeting of VGLUT3.

Threonine 8 is located in a region of the protein specific for VGLUT3 (in rodents and humans) and is not present in VGLUT1 or VGLUT2 (Fig. 2e). To gain further insight on the impact of the mutated allele in the brain, a mouse line where the Threonine in position 8 was mutated to an Isoleucine (T8I) was developed (Fig. 2g). The expression of VGLUT3 and VGLUT3-p.T8I were compared by immunoautoradiography and immunofluorescence on brain slices of WT mice and VGLUT3^T8I/T8I mice. As shown on Fig. 1h, the level of expression of the WT and the mutated alleles was comparable in all inspected brain areas. At the cellular level, both VGLUT3 isoforms were observed in sparse soma and proximal dendrites, as well as in an enriched plexus of varicosities in the hippocampus and the striatum (Fig. 2i–m). Therefore, the pT8i variant does not modify the quantitative or qualitative distribution of VGLUT3.

Finally, VAChT and VGLUT3 were visualized with STED microscopy in preparations of striatal synaptic vesicles from WT mice and VGLUT3^T8I/T8I mice. As previously reported[24], we observed a low-frequency overlay between VAChT and VGLUT3 immunofluorescent spots in both WT mice

and VGLUT3^T8I/T8I mice (Fig. 2n, o arrows). The average distance between the center of VAChT-immunopositive and VGLUT3-immonopositive fluorescent spots was estimated by a batch analysis of nearest neighbor distances (NNDs) in striatal synaptic vesicles from WT mice and VGLUT3^T8I/T8I mice. The frequency histogram revealed a first peak in a NND range between 0 and 120 nm (centered around a maximum at ≈50 nm) and a more spread-out low-frequency distribution for NNDs above 120 nm, for both WT mice and VGLUT3^T8I/T8I mice (Fig. 2p, Kolmogorov-Smirnov test, $p = 0.218$). A previous study determined that a threshold of 95 nm allows to identify synaptic vesicles expressing both VGLUT3 and VAChT from vesicles expressing either VGLUT3 alone or VAChT alone[24]. For both the WT and the mutated isoform, we found that ≈40% synaptic vesicles co-expressed both transporters, while approximately 60% of synaptic vesicles expressed either VAChT or VGLUT3 (Fig. 2q, Chi-squared, test $p = 0.439$). These observations suggest that the p.T8I mutation does not alter the relative distribution of VAChT and VGLUT3 in synaptic vesicles.

### Consequences of the p.T8I mutation on structural and molecular properties of VGLUT3

In VGLUT3, Threonine 8 is present in the cytoplasmic N-terminus (Fig. 3a). We explored whether the virtual 3D structure of VGLUT3 and

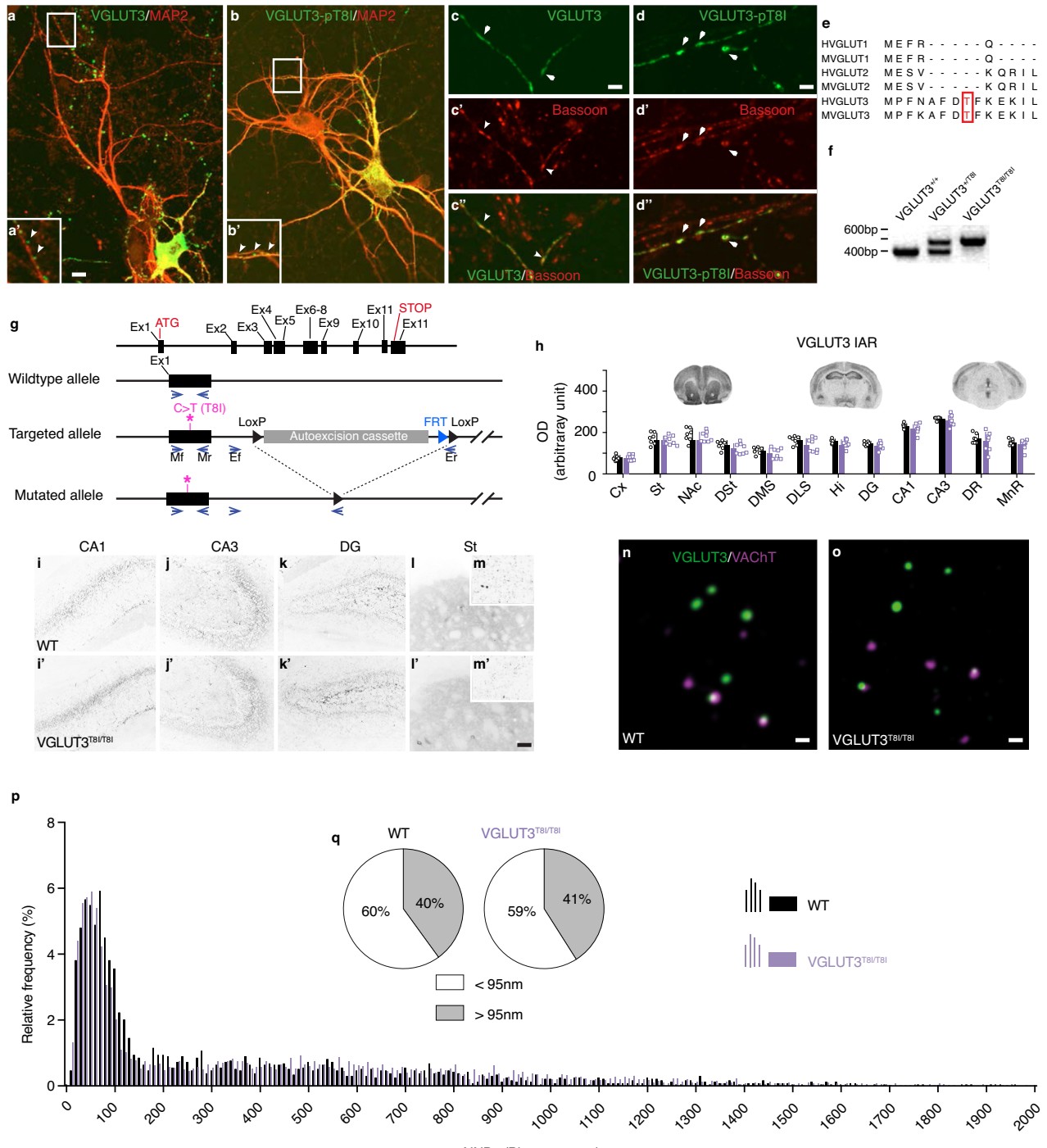

its N-terminus section were modified by the p.T8I variation. Using the E. coli D-galactonate:proton symporter (DgoT) for template[25], we observed that replacing Threonine 8 with Isoleucine had minimal impact on the overall structure of the transporter, as well as on the structure or positioning of its N-terminus (Fig. 3a–d). Importantly, in the presence or absence of Isoleucine 8, the N-terminus was located distantly form the transport pore (Fig. 3a–d). The putative 3D model predicts that the p.T8I variation should not have a major effect on glutamate vesicular accumulation operated by VGLUT3.

To confirm this prediction, we then assessed whether the p.T8I variant altered VGLUT3-dependent glutamatergic signaling. Synaptic vesicles prepared from the striatum of WT mice accumulated 330.3 ± 50 pmole.mg$^{-1}$ prot.10 min$^{-1}$ of [$^3$H]glutamate (Fig. 3e). This

glutamate transport represents the combined activity of VGLUT1, VGLUT2 and VGLUT3. In comparison, striatal synaptic vesicles from VGLUT3-null mice transported only 187.7 ± 34.7 pmole.mg$^{-1}$ prot.10 min$^{-1}$ of [$^3$H]glutamate (Fig. 3e, Kolmogorov-Smirnov test validated samples normality distribution, one-way ANOVA, p = 0.001; Tukey's post hoc test, $p$ = 0.005 WT mice vs VGLUT3-KO mice). This suggests that VGLUT3 accounted for ≈43% (143 pmol.mg$^{-1}$ prot.10 min$^{-1}$ [$^3$H]glutamate) of the total striatal vesicular uptake of glutamate. In the striatum VGLUT3 is present in all cholinergic varicosities which accounts for 15% striatal terminals and in a small subpopulation of serotoninergic varicosities[11,12,22]. Other glutamatergic terminals (originating from the cortex or the thalamus) can be estimated to account for ≈30–40% of total striatal glutamatergic

**Fig. 2 | The p.T8I variant does not alter the anatomical distribution of VGLUT3.**
**a-d**, Immunofluorescent detection of VGLUT3 (**a, a', c, c', c''**) or VGLUT3-p.T8I (**b, b', d, d', d''**) (green) and microtubule-associated protein (MAP2, **a, a', b, b'**) or bassoon (**c', c'', d', d''**) (red) in hippocampal neuronal culture of WT mice.
**e**, Alignment of human (H) or mouse (M) VGLUT1, VGLUT2 and VGLUT3 amino acid sequences. **f**, Genotyping of WT (VGLUT3$^{+/+}$) mice, heterozygous mice (VGLUT3$^{+/T8I}$) or homozygous mice (VGLUT3$^{T8I/T8I}$). **g**, Schematic representation of the targeting strategy and genotyping strategy of mouse VGLUT3 by PCR. Mice were genotyped with primers (arrows) flanking exon 1 (mf, mr, ex1), in intron 1 (ef) and in the LoxP sites in 3' of the autoexcision cassette (er). Asterisks represent the targeted site in exon 1. **h**, Top: Immunoautoradiographic (IAR) regional distribution of VGLUT3 and VGLUT3-p.T8I on coronal sections from the brain of WT mice (black, n = 7 mice) and VGLUT3$^{T8I/T8I}$ mice (purple, n = 8 mice). Bottom: Densitometric quantification of VGLUT3 and VGLUT3-p.T8I on mouse brain sections. Data are presented as mean values ± SEM. Statistical analysis performed with two-sided unpaired t-test. **i-m'**,
Immunofluorescent detection of VGLUT3 (**i-m**) or VGLUT3-p.T8I (**i'-m'**) in the hippocampus and in the striatum of WT mice or VGLUT3$^{T8I/T8I}$ mice. **n-o**, Immunofluorescent detection with STED microscopy of VGLUT3 or VGLUT3-p.T8I (purple) and VAChT (green) in preparations of striatal synaptic vesicles. **p**, Automatic batch analysis of nearest neighbor distances (NNDs) between VGLUT3- or VGLUT3-p.T8I- and VAChT-immunofluorescent spots on striatal isolated synaptic vesicles. Statistical analysis two-sided Kolmogorov-Smirnov test, p = 0.218. **q**, Percentage of VGLUT3 or VGLUT3-p.T8I immunofluorescent spots displaying a NND with their closest VAChT-positive spot above and below 95 nm (p = 0.439, two-sided Chi-squared test). Source data are provided as a Source Data file. Abbreviations: CA1-3, fields of the hippocampus pyramidal layer; Cx, cortex; DG, dentate gyrus; DMS, dorsomedial striatum; DLS, dorsolateral striatum; DR, dorsal raphe nucleus; st, striatum; DSt, dorsal striatum; Hi, hippocampus; IAR, immunoautoradiography; MnR, median raphe nucleus; NAc, nucleus accumbens. Bar: 10 μm in **a** and **b**, 5 μm in **a'** and **b'**, 2 μm in **c-d''**, 10 μm in **i-l'**, 0.1 μm in **n** and **o**.

terminals. It is therefore not surprising that VGLUT3 contributes to ≈40% of the total [³H] glutamate vesicular accumulation in the striatum.

Strikingly, glutamate accumulation by synaptic vesicles isolated from the striatum of VGLUT3$^{T8I/T8I}$ mice (350.7 ± 38.7 pmole.mg$^{-1}$ prot.10 min$^{-1}$ [³H]glutamate) was not different from the one observed with synaptic vesicles obtained from the striatum of WT mice (Fig. 3e, Kolmogorov-Smirnov test validated sample normality distribution, one-way ANOVA, p = 0.001; Tukey's post hoc test, p = 0.851). This finding indicates that the p.T8I allele does not influence glutamate vesicular accumulation catalyzed by VGLUT3.

To explore the effect of p.T8I on glutamate release, we then used electrophysiological recordings of isolated hippocampal neurons from VGLUT1-KO newborn mice rescued by either VGLUT3 or VGLUT3-p.T8i (Fig. 3f–h)[26]. Analysis of spontaneous release activity demonstrated that mean miniature excitatory postsynaptic currents (mEPSC) amplitude (Fig. 3f, Dunn's post hoc test, p > 0.999), frequency (Fig. 3g, Dunn's post hoc test, p > 0.999), or charge (Fig. 3h, Dunn's post hoc test, p > 0.999), were similar in autapses rescued by VGLUT3 or by VGLUT3-p.T8I. These results demonstrate that the p.T8I mutation does not change the amount of glutamate release, the cycling of the ready-releasable pool (RRP) of vesicles or the release probability.

One additional important basic property of VGLUT3 in ChIs is its ability to enhance ACh vesicular accumulation[11]. Striatal vesicles from WT mice accumulated 11.9 ± 1.2 pmole.mg$^{-1}$ prot.10 min$^{-1}$ of [³H]ACh under basal conditions and 25.3 ± 2.5 pmole.mg$^{-1}$ prot.10 min$^{-1}$ of [³H]ACh in the presence of 1 mM glutamate (Fig. 3i). This +113% increase (Bonferroni's post hoc test, p < 0.001) of [³HACh accumulation in presence of glutamate was previously shown to be due to VGLUT3-dependent vesicular synergy[11]. In VGLUT3$^{T8I/T8I}$ mice, [³H]ACh vesicular uptake was also increased by glutamate (Fig. 3i, basal: 9.27 ± 1.15 pmole.mg$^{-1}$ prot.10 min$^{-1}$; with 1 mM glutamate: 16.21 ± 0.68, Bonferroni's post hoc test, p = 0.044). The magnitude of the increase ( +74%) was lower than the one observed with striatal vesicles from WT mice. Therefore, vesicular [³H]ACh accumulation under basal conditions was comparable between WT mice and VGLUT3$^{T8I/T8I}$ mice (Bonferroni's post hoc test, p > 0.999) whereas [³H]ACh uptake in presence of 1 mM glutamate was higher in WT mice compared to mutant mice (Bonferroni's post hoc test, p = 0.003). These data show that vesicular synergy is partially blunted by the p.T8I mutation, and that the amount of ACh stored in synaptic vesicles from ChIs might be reduced in VGLUT3$^{T8I/T8I}$ mice.

These observations suggest that ACh release may be decreased in the striatum of VGLUT3$^{T8I/T8I}$ mice compared to WT mice. To test this hypothesis, we performed fiber photometry with ACh biosensor GACh4.3 expressed in the dorsomedial striatum (DMS) of WT mice and VGLUT3$^{T8I/T8I}$ mice[27,28] (Fig. 3j–n). Spontaneous ACh release events were recorded in freely moving animals placed in an open field. Importantly,

in these recordings, both the median (Fig. S2a, unpaired t-test, p = 0.868) and the threshold were comparable between WT mice and VGLUT3$^{T8I/T8I}$ mice (Fig. S2b, unpaired t-test, p = 0.584). We found that while the amplitude of cholinergic events was unchanged (Fig. 3l, Wilcoxon test, p = 0.382), the frequency of events was on average two times lower in VGLUT3$^{T8I/T8I}$ mice compared to WT mice (Fig. 3m, unpaired t-test p = 0.009, Wilcoxon rank sum test with continuity correction (unpaired), W = 43.5, p-value = 0.017). Accordingly, the distribution of inter-event intervals (IEIs) was different between WT mice and VGLUT3$^{T8I/T8I}$ mice, with an increase in the proportion of long IEIs in mutant mice (Fig. 3n, Kolmogorov-Smirnov test, p = 0.001). These results demonstrate that, subsequently to the reduction of vesicular synergy, basal ACh transmission is reduced in VGLUT3$^{T8I/T8I}$mice.

ACh-glutamate co-transmission from ChIs exerts a dual control over DA release in the DMS but not in dorsolateral striatum (DLS)[15,17,29,30]. In line with this model, we predicted that mice with a decreased ACh striatal tone should exhibit reduced DA transmission in the DMS but not in the DLS. Using a quantitative approach such as in vivo voltammetry, we indeed observed that KCl-induced DA release was significantly decreased in the DMS of VGLUT3$^{T8I/T8I}$ mice compared to controls (−31%, Fig. 3o, p, unpaired t-test p = 0.03), whereas it was not different in the DLS of WT mice and mutant mice (Fig. 3q, r, unpaired t-test, p = 0.849).

These results demonstrate that VGLUT3$^{T8I/T8I}$ mice have reduced cholinergic tone in the striatum and a lower DA release in the DMS compared to the DLS.

## VGLUT3$^{T8I/T8I}$ mice display no alteration of locomotor activity or anxiety

We then inspected behavioral consequences of these neurochemical changes. It was previously found that mice with a loss of VAChT in ChIs (VAChTcKO mice) have normal basal locomotor activity, whereas VGLUT3 ablation results in increased basal locomotor activity in novel environments or during the awake cycle[11,31,32]. We recorded horizontal activity of WT mice and VGLUT3$^{T8I/T8I}$ mice over a full 24 h day-night cycle (Fig. 4a). Both WT mice and VGLUT3$^{T8I/T8I}$ mice demonstrated a similar profile of increased ambulation during the awake cycle (night) (two-way RM ANOVA, genotype x time, F$_{30,420}$ = 0.794, p = 0.776). This result further supports the idea that VGLUT3$^{T8I/T8I}$ mice have a loss of cholinergic signaling and are more comparable to VAChTcKO mice than to VGLUT3-KO mice.

Loss of VGLUT3 or loss of VAChT function both results in increased levels of anxiety[16,33,34]. In the elevated plus maze, VGLUT3$^{T8I/T8I}$ mice and WT mice made similar number of entries into the closed (unpaired t-test, p = 0.926) and open arms (unpaired t-test, p = 0.632), spent similar amounts of time in the open arms and showed a comparable number of transitions between closed and open arms (Fig. 4b–d). Similar results were obtained in the O-maze (Fig. S3). We

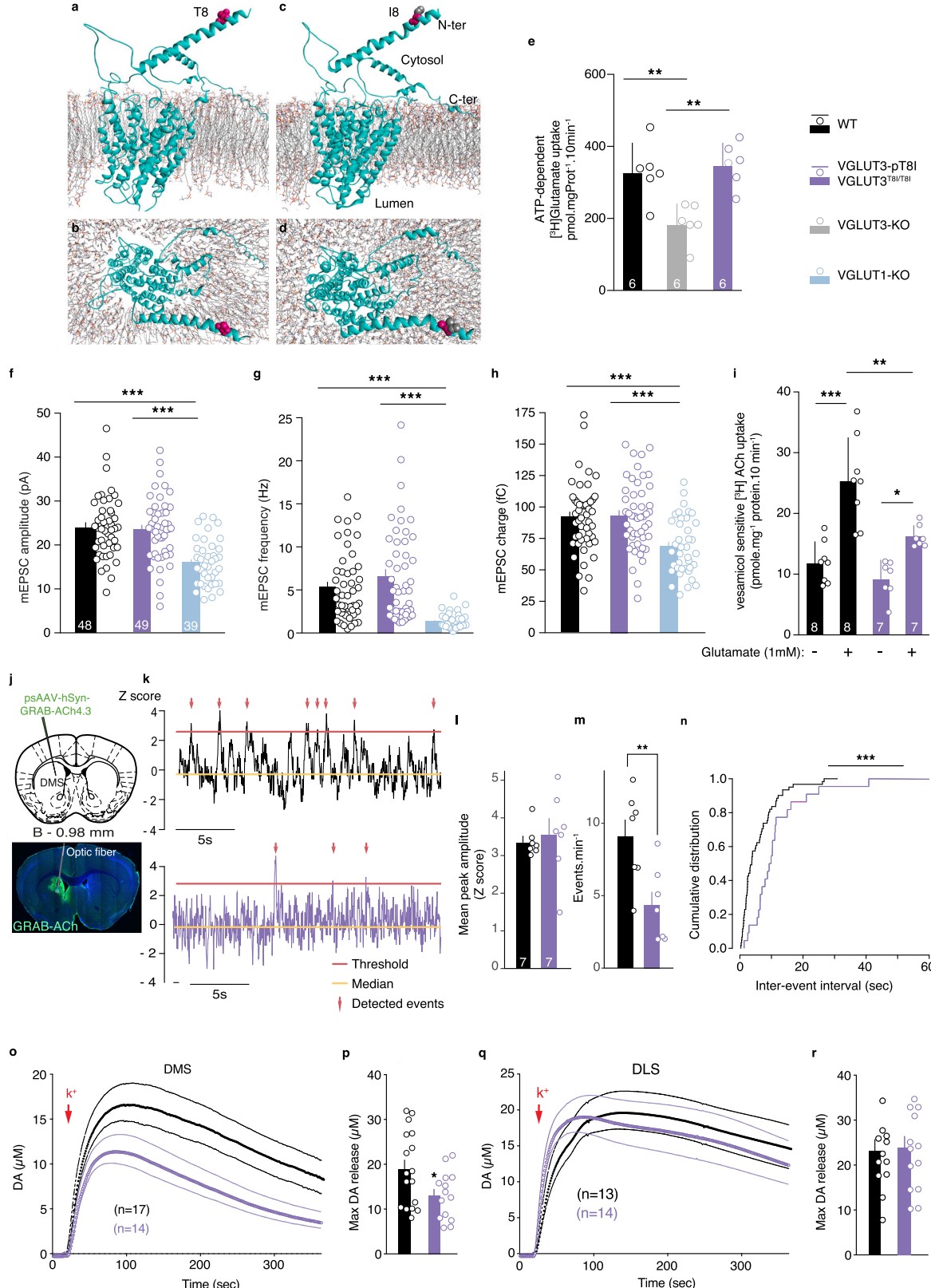

also used marble burying test in a novel environment to assess anxiety induced by neophobia and repetitive behaviors. After 10 min in the novel environment, WT mice and VGLUT3$^{T8I/T8I}$ mice buried a similar number of marbles (Fig. 4e). Taken together, results from EPM, O-maze, and marble burying tests suggest that VGLUT3$^{T8I/T8I}$ mice do not exhibit excessive levels of anxiety.

## VGLUT3$^{T8I/T8I}$ mice exhibit increased cue-induced reinstatement of cocaine seeking

Reinforcing properties of cocaine were evaluated in VGLUT3$^{T8I/T8I}$ mice using operant self-administration. Mice were first trained in 5 sessions of fixed-ratio 1 (FR1) followed by 5 sessions of FR3 (Fig. 4f). During FR1 and FR3 sessions, the number of nosepokes on the inactive lever was

**Fig. 3 | The VGLUT3-p.T8I variant does not affect vesicular accumulation and release of glutamate but impairs ACh and DA signaling in the dorsal striatum.**
**a**, Three-dimensional model of side and top view of VGLUT3 (**a,b**) and VGLUT3-p.T8I (**c,d**). **e**, Glutamate vesicular uptake in striatal vesicles from WT mice, VGLUT3$^{T8I/T8I}$ mice or VGLUT3-KO mice (for each genotype, n = 6 independent determinations). **$p$ = 0.005 for WT mice vs VGLUT3-KO mice and **$p$ = 0.002 WT mice vs VGLUT3$^{T8I/T8I}$ mice, one-way ANOVA, Tukey's test post hoc analysis. **f-h**, Electrophysiological recordings of VGLUT1-KO autaptic neurons expressing VGLUT3 (WT) or VGLUT3-p.T8I. Plots of mean amplitude (**f**) frequency (**g**) or charge (**h**) of responses of autaptic neurons. ***$p$ < 0.001 WT mice (n = 48 independent autapses recording) *vs* VGLUT3$^{T8I/T8I}$ mice (n = 49 independent autapses recording) vs VGLUT1-KO mice (n = 39 independent autapses recording), Kruskal-Wallis test; Dunn's test post hoc analysis VGLUT3$^{T8I/T8I}$ mice vs VGLUT1-KO mice and WT mice vs VGLUT1-KO mice. **i**, Vesicular acetylcholine uptake measured in striatal synaptic vesicles, in absence (-) or presence (+) of glutamate (1 mM). ***$p$ < 0.001 WT mice / Glu- *vs* WT mice / Glu + ; *$p$ = 0.044 VGLUT3$^{T8I/T8I}$ mice / Glu- *vs* VGLUT3$^{T8I/T8I}$ mice /

Glu + ; **$p$ = 0.003 WT mice / Glu+ *vs* VGLUT3$^{T8I/T8I}$ mice / Glu + , (two-way ANOVA, Bonferroni's test post hoc analysis) (Glu- n = 8 independent determinations, Glu + = n = 7 independent determinations). **j**, AAV-mediated delivery of GRAB-ACh4.3 sensor in DMS. **k**, Examples of fiber photometry recording (ΔF/F) for WT mice (black) or VGLUT3$^{T8I/T8I}$ mice (purple). **l, m** Mean peak amplitude and frequency of spontaneous ACh events (n = 7 WT mice and n = 7 VGLUT3$^{T8I/T8I}$ mice). **$p$ = 0.009 (two-sided unpaired t-test) and Wilcoxon rank sum test with continuity correction (non-paired), W = 43.5, $p$ = 0.017. **n**, Cumulative distribution of inter-event intervals. **$p$ = 0.001 (two-sided Kolmogorov-Smirnov test). **o-r**, In vivo voltammetry of DA K$^+$-evoked release in DMS (**o**) or DLS (**q**) of WT mice (black) and VGLUT3$^{T8I/T8I}$ mice (purple). (**o**) Genotype, $F_{1,29}$ = 5.63, $p$ = 0.024; time, $F_{2,499, 72.46}$ = 39.76, $p$ < 0.001; genotype x time, $F_{399, 11571}$ = 1.84 $p$ < 0.001 (two-way RM ANOVA, Bonferroni's test post hoc analysis). Maximum level of DA release in DMS (**p**, n = 17 WT mice and n = 14 VGLUT3$^{T8I/T8I}$ mice), or DLS (**r**, n = 13 WT mice and n = 14 VGLUT3$^{T8I/T8I}$ mice). (**p**) *p = 0.03 (two-sided unpaired t-test). Data are presented as mean values ± SEM in **e, f-i, l, m, p, r**. Source data are provided as a Source Data file. Bar: 1000 μm in **j**.

---

similarly minimal in both genotypes (Fig. S3f). No statistical differences were obtained in active nose-pokes between genotypes (Fig. 4f, two-way RM ANOVA, genotype $F_{1,23}$ = 1.641, $p$ = 0.2129). Animals discriminated between the active and inactive lever similarly above 75%. Specifically, WT mice showed discrimination of 81.92 ± 2.74 during FR1 and 89.31 ± 2.15 during FR3. Similarly, mutants showed a level of discrimination of 79.19 ± 3.38 in FR1 and 86.61 ± 3.54 in FR3. VGLUT3$^{T8I/T8I}$ mice showed a non-significant trend to perform less active nosepokes during FR1 but they reached similar levels of responding as WT mice after 10 sessions of training in FR1 and FR3. Concerning the number of infusions in FR1 and FR3, VGLUT3$^{T8I/T8I}$ mice showed a significant trend to perform less than WT mice (Fig. S3g, genotype x time, $F_{9,207}$ = 2.085, $p$ < 0.05, two-way RM ANOVA). Particularly, post-hoc analysis showed that in sessions 3 and 4 of FR1, VGLUT3$^{T8I/T8I}$ mice obtained significantly fewer cocaine infusions than WT mice (Fig. S3g, least significant difference (LSD) post hoc test day 3 *$p$ < 0.05 and day 4 **$p$ < 0.01). However, the number of infusions and the amount of cocaine obtained by WT mice and VGLUT3$^{T8I/T8I}$ mice were similar during final sessions of the FR3 training (Fig. S3g). Interestingly, the percentage of mice reaching criteria of operant conditioning learning was 71 % for WT mice and 36% for VGLUT3$^{T8I/T8I}$ mice following FR1 and FR3 training (Fig. 4g, chi-squared test = 6.627, $p$ < 0.05). Only mice reaching the acquisition criteria were subsequently tested in a progressive ratio (PR) session followed by 22 days of extinction and subsequent cue-induced reinstatement session (Fig. 4h–j). In the PR paradigm, breaking points were not significantly different between WT mice (37.43 ± 8.01, Fig. 4h) and VGLUT3$^{T8I/T8I}$ mice (31.45 ± 7.78). As shown in Fig. 4i, the extinction of cocaine self-administration followed a similar progressive decay in WT mice and VGLUT3$^{T8I/T8I}$ mice. After extinction procedure, mice were evaluated for cue-induced reinstatement of cocaine-seeking behavior. When reaching the extinction criterion, the cumulated number of active nosepoking responses was significantly lower than the responses during the acquisition of self-administration criteria (Fig. 4j, paired t-test, $p$ < 0.01 for WT mice and paired t-test, $p$ < 0.05 for VGLUT3$^{T8I/T8I}$ mice). During the reinstatement test, the cumulated number of active nosepoking responses was significantly higher than that obtained on the day of achieving the extinction criterion (Fig. 4j left panel, paired t-test, $p$ < 0.01 in WT mice and $p$ < 0.01 for VGLUT3$^{T8I/T8I}$ mice). All mice reached the same or higher level of responses than the one acquired during acquisition training. Additionally, the behavioral pattern of operant response during the last extinction session and the cue-induced reinstatement test were dissected in 10 min blocks to explore differential signatures between genotypes (Fig. S3h, i). Results showed a similar pattern between genotypes in the last extinction session (Fig. S3h) but a differential pattern of operant behavior during the cue-induced reinstatement (Fig. S3i). VGLUT3$^{T8I/T8I}$ mice responded more in the second half of the session compared to WT and compared to the previous last

extinction session in minutes 70–100 (genotype x time, $F_{11,253}$ = 2.6284, $p$ < 0.01, two-way RM ANOVA, and LSD post hoc test 70 min ***$p$ < 0.001, 80 min ***$p$ < 0.001, 90 min ***$p$ < 0.05 and 100 min **$p$ < 0.05, WT mice vs VGLUT3$^{T8I/T8I}$ mice). Interestingly, both genotypes showed distinct behavioral signatures in cue-induced reinstatement.

Therefore, despite the fact that mutant mice tend to self-administer less cocaine in FR1, following forced extinction, VGLUT3$^{T8I/T8I}$ mice are more vulnerable to cocaine-seeking relapse induced by exposure to drug-associated cues, a central feature of SUDs. This finding supports the notion that the p.T8I allele may be causal in increased vulnerability to cocaine abuse.

## VGLUT3$^{T8I/T8I}$ mice show facilitated habit formation and vulnerability to maladaptive eating

VAChTcKO mice are hypocholinergic in the striatum and have a decreased ability to interrupt habitual behaviors[15]. Based on the finding that VGLUT3$^{T8I/T8I}$ mice also exhibit decreased striatal cholinergic tone, we next explored a potential bias toward habits in VGLUT3$^{T8I/T8I}$ mice. Mice were trained for 16 consecutive days of FR1 schedule for sucrose pellets. During operant instrumental training, WT mice and VGLUT3$^{T8I/T8I}$ mice demonstrated identical nosepoke activity to obtain sucrose pellets (Fig. 5a, two-way RM ANOVA, genotype $F_{1,22}$ = 0.0001, $p$ = 0.992). As expected after FR1 training, WT mice showed a marked decrease of active nosepokes in devalued condition compared to valued condition (valued: 42.8 ± 6,5, devalued: 21.0 ± 5.2, paired t-test $p$ = 0.001, Fig. 5b), demonstrating that seeking for sucrose in WT mice was a goal-directed behavior. In contrast, VGLUT3$^{T8I/T8I}$ mice exhibited similar performances in both conditions (Fig. 5b valued: 36.3 ± 5.95, devalued: 36.8 ± 9.5, paired t-test $p$ = 0.945), indicating that after only 16 days of FR1 training mutant mice already switched to habits (Fig. 5b, two-way RM ANOVA, Value $F_{1,36}$ = 54.11, $p$ = 0.0131; Genotype x Value $F_{123,3036}$ = 1.43, $p$ = 0.0102). These data suggest that, compared to WT mice, VGLUT3$^{T8I/T8I}$ mice have a bias towards habit formation.

Since mutant mice demonstrate a vulnerability to reinstate cocaine-seeking behavior, we wondered whether food addiction could be a confounding factor in the above-described test. We therefore explored the behavior of VGLUT3$^{T8I/T8I}$ mice in a recently developed mouse model of food addiction[35]. WT mice and VGLUT3$^{T8I/T8I}$ mice underwent chocolate-flavored pellets self-administration in 6 FR1 sessions, followed by FR5 schedule for 118 sessions. WT mice and mutant mice demonstrated similar and stable nosepoking under FR1 and FR5 (Fig. 5c, mixed effect model (restricted maximum likelihood, REML), genotype $F_{1,25}$ = 0.059, $p$ = 0.811). This result suggests that natural rewards, like palatable pellets, are not more reinforcing for mutant mice than for WT mice. We also observed in the two-bottle choice that preference for the sucrose solution was comparable in WT mice and in VGLUT3$^{T8I/T8I}$ mice (Fig. S3a, unpaired t-test $p$ = 0.341).

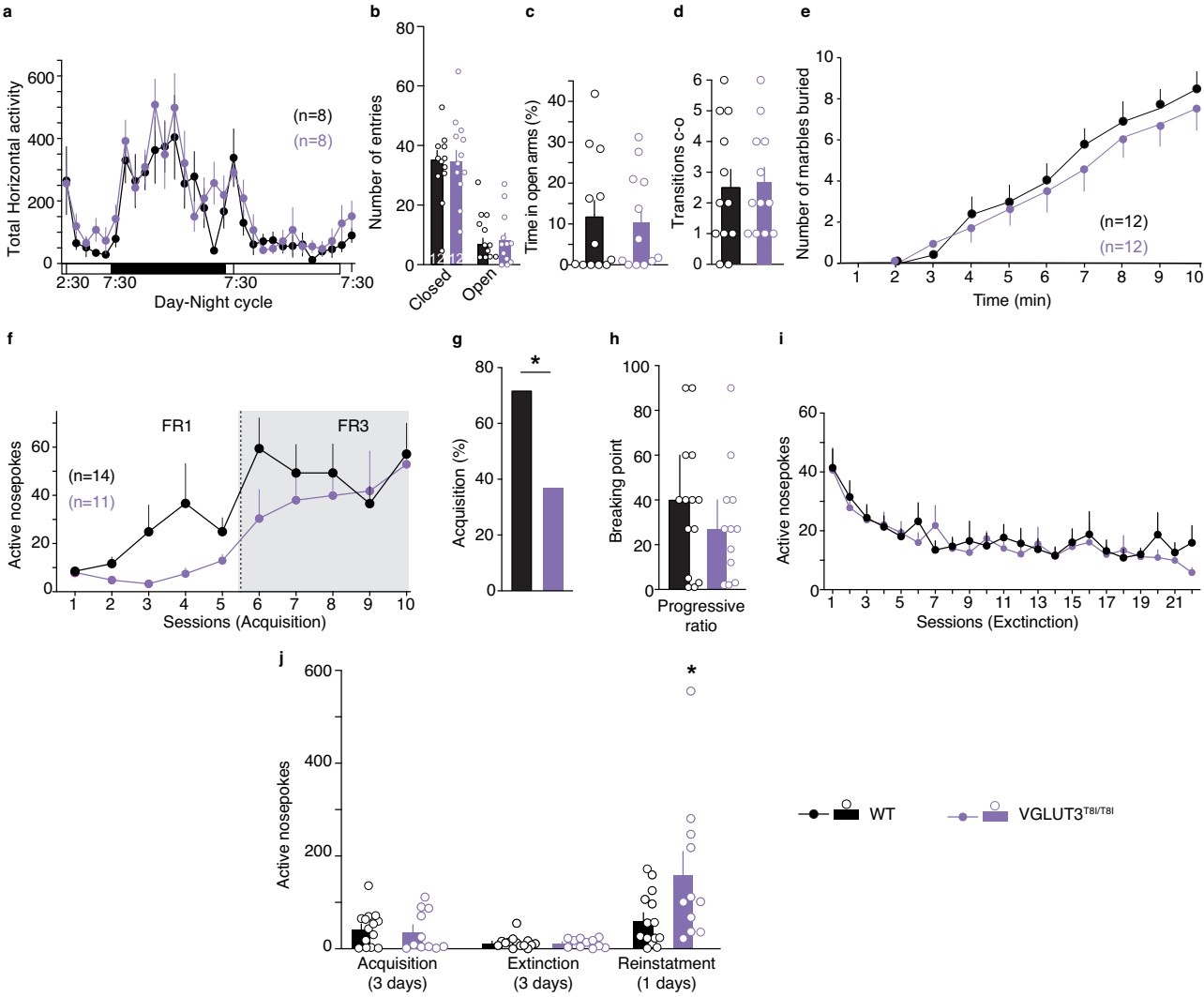

**Fig. 4 | VGLUT3^(T8I/T8I) mice exhibit normal locomotor activity and anxiety phenotype but show increased vulnerability to cue-induced reinstatement of cocaine seeking compared to controls. a**, Horizontal locomotor activity of WT mice (black) or VGLUT3^(T8I/T8I) mice (purple) measured during a complete day-night cycle ($p = 0.776$, two-way RM ANOVA). **b-d**, Elevated plus maze (n = 12 WT mice and n = 12 VGLUT3^(T8I/T8I) mice). **b**, Number of entries in closed or open arms **c**, Time spent in open arms versus closed arms (%). **d**, Number of transitions from closed (c) to open (o) arms. **e**, Marble-burying test. Statistical analyses were performed with two-way RM ANOVA and post hoc comparison with Bonferroni's test (**a,e**) and two-sided unpaired t-test (**b-d**). **f-j**, Cocaine (0.5 mg.kg⁻¹ per infusion, iv) self-administration in WT mice (black, n = 14 mice) or VGLUT3^(T8I/T8I) mice (purple, n = 11 mice). **f**, Number of active nosepokes during the acquisition (fixed ratio 1 (FR1) and 3 (FR3)) of self-administration. genotype x time $F_{9,207} = 0.972$, $p = 0.464$, two-way RM ANOVA. **g**, Percentage of mice reaching the criteria for operant conditioning learning. * $p = 0.010$ (two-sided chi-squared test). **h**, Motivation for cocaine measured by the breaking point achieved in the progressive ratio schedule of reinforcement (two-sided unpaired t-test, $p = 0.60$). **i**, Number of active nosepokes during extinction procedure (genotype x time, $F_{21,483} = 0.7128$, $p = 0.8215$, two-way RM ANOVA). **j**, Cue-induced reinstatement with the mean number of active nosepokes during the different experimental phases: acquisition of cocaine self-administration behavior (mean of 3 days acquisition criteria), extinction (mean of 3 days extinction criterion) and reinstatement induced by cues. Number of active nosepokes during cue-induced reinstatement test (two-sided unpaired t-test, * $p = 0.041$). Data are presented as mean values ± SEM in **a-j**. Source data are provided as a Source Data file.

Following operant training (FR1/FR5, Fig. 5c), three behavioral tests were used to evaluate the food addiction-like phenotype[35]. No significant difference was observed between the two genotypes in the three criteria: persistence to response evaluated by the number of active responses during the pellet-free period (Fig. 5d, non-parametric Mann-Whitney test U = 67, $p = 0.2088$), motivation evaluated in the progressive ratio test (Fig. 5e, unpaired t-test $p = 0.677$), and compulsivity evaluated by pairing pellet obtention with punishment (Fig. 5f, unpaired t-test $p = 0.597$). Following these tests, only one VGLUT3^(T8I/T8I) mouse had 3 criteria whereas no WT mice had all 3 criteria, 21 % of WT mice and 23% of VGLUT3^(T8I/T8I) mice had 2 criteria and could be considered as "addicted" to food, whereas 78% of WT mice and 77% of VGLUT3^(T8I/T8I) mice had at most 1 criterion and were classified as "non-addicted" animals (Fig. 5g and h, respectively) (Chi-squared test

addicted vs non addicted, and VGLUT3^(T8i/T8i) mice vs WT mice $p = 0.918$. Furthermore, significant Pearson positive correlations were observed between the number of addiction criteria met and the level of responses for each criterion in WT mice, but only between criteria and compulsivity in VGLUT3^(T8I/T8I) mice (Fig. 5i–k, statistics in Supplementary Table 5). Therefore, the p.T8I allele facilitates habit formation and increases the vulnerability to cocaine relapse without altering the vulnerability to food addiction.

Importantly, a patient carrying the p.T8I allele presented clinically both SUDs (cocaine addiction) and EDs (bulimia nervosa). We therefore assessed the vulnerability of VGLUT3^(T8I/T8I) mice to maladaptive eating in two well-established mouse models reminiscent of bulimia nervosa (binge-like sucrose overconsumption test) and anorexia nervosa (activity-based anorexia (ABA) model)[36,37].

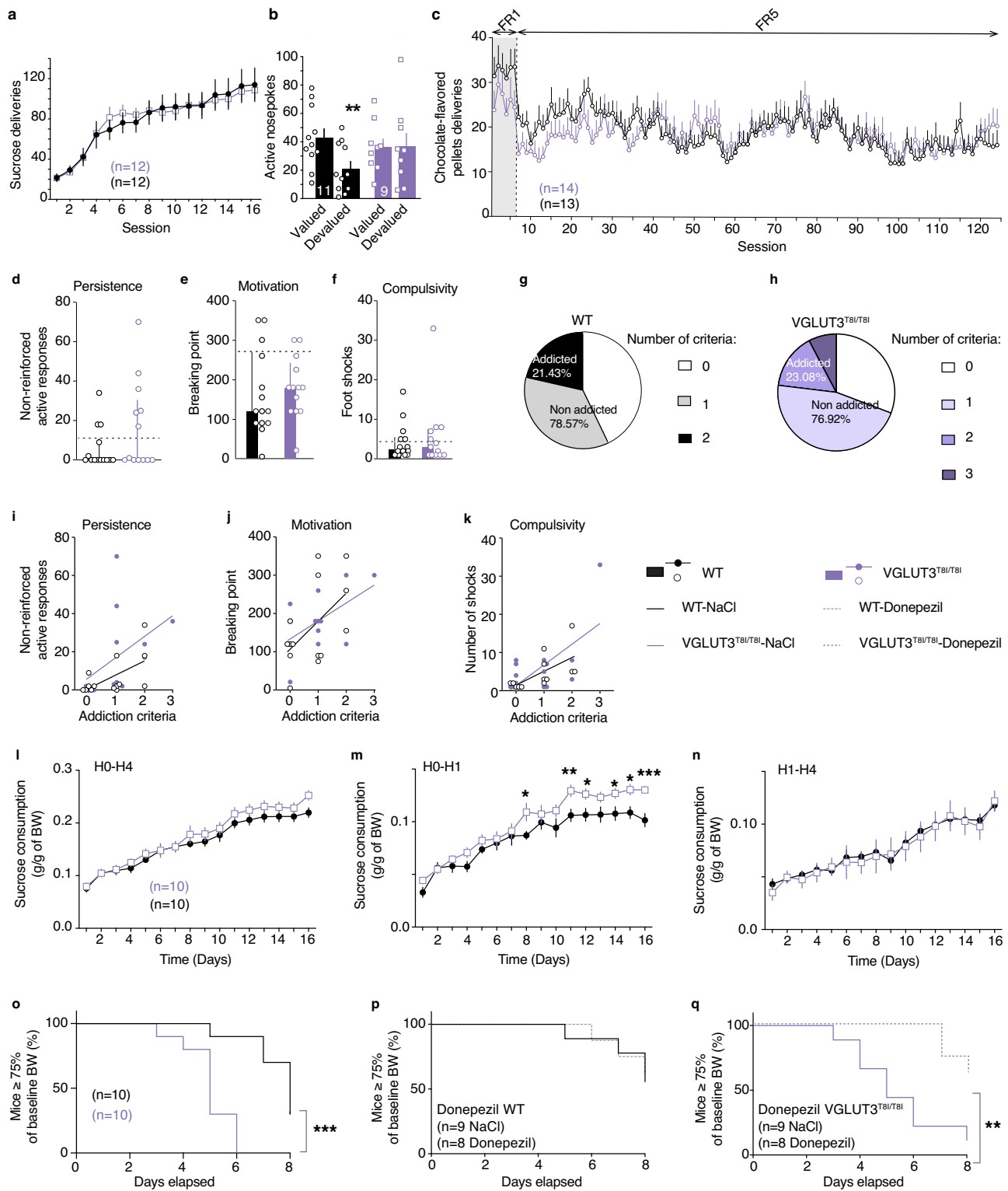

During the 16 days of the binge-like sucrose overconsumption test[37], WT mice and VGLUT3[T8I/T8I] mice consumed similar amount of highly concentrated sucrose solution during the 4 h of the test (Fig. 5l, two-way RM ANOVA, genotype $F_{1,18} = 0.96$, $p = 0.34$,). In contrast, after 7 days, the sucrose consumption of mutant mice increased to a higher extent during the first hour of access compared to controls (H0-H1, Two-way RM ANOVA, genotype: $F_{15,270} = 3.066$, $p < 0.001$, Fig. 5m). Sucrose intake was similar in WT mice and mutant mice during the final 3 h of the test (Fig. 5n H1-H4, two-way RM ANOVA, genotype: $F_{1,18} = 0.027$, $p = 0.872$,). As shown by the two-bottle choice, increased

binge-like consumption of sucrose during the first hour of the test in VGLUT3[T8I/T8I] mice was not due to an increased sucrose preference (Fig. S3a, unpaired t-test $p = 0.341$). Water and food intake were similar in both genotypes during the 16 days of the test (Fig. S3b, c, water intake, two-way RM ANOVA, genotype: $F_{1,18} = 0.828$, $p = 0.378$; Fig. S3b, food intake two-way RM ANOVA, genotype: $F_{1,18} = 0.128$, $p = 0.725$). Moreover, excessive sucrose consumption during the first hour (H0-H1) by VGLUT3[T8I/T8I] mice compared to WT mice was still observed after exposing the animals to food ad libitum for 1 h (Fig. S4d chow preload, unpaired t-test $p < 0.001$). Following chow preload, WT mice and

**Fig. 5 | VGLUT3$^{T8I/T8I}$ mice are more prone to habit formation and display an increased vulnerability to develop maladaptive eating compared to WT mice.** **a**, Number of sucrose pellets deliveries for VGLUT3$^{T8I/T8I}$ mice (purple) versus controls (black) during FR1 training (genotype, $F_{1,22} = 0.0001$, $p = 0.992$; time $F_{3.064,67.41} = 54.11$, $p < 0.001$; genotype x time $F_{15,330} = 0.548$, $p = 0.912$; two-way RM ANOVA and post hoc comparison with the method of contrasts). **b**, WT mice/valued *vs* WT mice/devalued **$p = 0.001$, two-sided paired t-test ($n = 11$ mice). VGLUT3$^{T8i/T8i}$ mice/valued *vs* VGLUT3$^{T8I/T8I}$ mice/devalued $p = 0.945$, two-sided paired t-test ($n = 9$ mice). **c**, Number of chocolate-flavored pellets deliveries during FR1 and FR5 training (genotype $F_{1,25} = 0.059$, $p = 0.811$; time $F_{9,073,223.9} = 5.468$, $p < 0.001$; genotype x time $F_{123,3036} = 1.43$, $p = 0.002$; Mixed effect model (REML)). **d**, Persistence to response ($p = 0.113$, two-sided unpaired t-test). **e**, Motivation ($p = 0.677$, unpaired t-test). **f**, Compulsivity ($p = 0.597$, two-sided unpaired t-test). Percentage of WT mice (**g**) and VGLUT3$^{T8I/T8I}$ mice (**h**) categorized as addicted or non-addicted ($p = 0.918$, Chi-squared test). **i-k**, Pearson correlations between individual values of addiction-like criteria and persistence (WT mice $r^2 = 0.334$, $p = 0.031$; VGLUT3$^{T8I/T8i}$ mice $r^2 = 0.2$, $p = 0.126$) (**i**), motivation (WT mice $r^2 = 0.295$, $p = 0.045$; VGLUT3$^{T8I/T8I}$ mice $r^2 = 0.305$, $p = 0.051$) (**j**) and compulsivity (WT mice $r^2 = 0.414$, $p = 0.013$; VGLUT3$^{T8I/T8I}$ mice $r^2 = 0.405$, $p = 0.011$) (**k**). **l-n**, Sucrose binge-like overconsumption test ($n = 10$ WT mice and $n = 10$ mice for VGLUT3$^{T8I/T8I}$ mice). Daily sucrose consumption during 4 h (H0-H4) (genotype $F_{1,18} = 0.96$, $p = 0.34$; time $F_{15,270} = 164.8$, $p < 0.001$; genotype x time $F_{15,270} = 1.432$, $p = 0.132$, Two-way RM ANOVA) (**l**), First hour (H0-H1) (genotype $F_{1,18} = 3.692$, $p = 0.071$; time $F_{5.587,100.6} = 119.5$, $p = <0.001$; genotype x time $F_{15,270} = 3.066$, $p < 0.001$, Two-way RM ANOVA) (**m**) or last 3 h (H1-H4) (genotype $F_{1,18} = 0.027$, $p = 0.872$; time $F_{15,270} = 42.37$, $p < 0.001$; genotype x time $F_{15,270} = 3.066$, $p < 0.001$) (**n**) of access to sucrose solution. **o-q**, ABA model ($n = 10$ WT mice and $n = 10$ VGLUT3$^{T8I/T8I}$ mice). **o**, Survival curve (Log-rank (Mantel-Cox) post hoc comparison $p < 0.001$, Gehan-Breslow-Wilcoxon post hoc comparison $p < 0.001$, Kaplan-Meier test). **p-q** Effect of chronic donepezil (0.3 mg.kg$^{-1}$) treatment on survival curves (Log-rank (Mantel-Cox) **p**ost hoc comparison $p = 0.819$, Gehan-Breslow-Wilcoxon post hoc comparison $p = 0.83$, Kaplan-Meier test) (**p**) or VGLUT3$^{T8I/T8I}$ mice (Log-rank (Mantel-Cox) post hoc comparison $p = 0.006$, Gehan-Breslow-Wilcoxon **p**ost hoc comparison $p = 0.004$, Kaplan-Meier test) (**q**). Data are presented as mean values ± SEM in a-f, i-n. Source data are provided as a Source Data file.

VGLUT3$^{T8I/T8I}$ mice displayed no difference of sucrose consumption during the final 3 h of the test (H1-H4). Therefore, the significant difference observed in H0-H4 (Fig. S4d chow preload, unpaired t-test, $p = 0.037$) between the two genotypes is driven solely by the overconsumption of VGLUT3$^{T8I/T8I}$ mice during the first hour of sucrose access.

Importantly, the quantity of food consumed during this 1 h ad libitum food exposure was similar in both mice groups (Fig. S4e, unpaired t-test, $p = 0.179$), ensuring that the hunger level of WT mice and mutant mice was comparable. The rapid ingestion of high-caloric rewarding food without physical hunger is reminiscent of a central feature of binge eating observed in bulimia nervosa. This finding suggests that the p.T8I allele could be the cause of an increased vulnerability to binge eating.

We next evaluated the vulnerability of VGLUT3$^{T8I/T8I}$ mice to self-starvation in the ABA model[36]. During the habituation phase, in the presence of the wheel (7 days), WT mice and VGLUT3$^{T8I/T8I}$ mice showed no difference in food intake or body weight (Fig. S4f, food intake two-way RM ANOVA, genotype: $F_{1,18} = 0.02$, $p = 0.887$; Fig. S4g, body weight, two-way RM ANOVA, genotype: $F_{1,18} = 0.041$, $p = 0.841$). In contrast, during the 3 final days (days 6–8) of the food restriction phase, VGLUT3$^{T8I/T8I}$ mice displayed a greater reduction of food intake and body weight than control mice (Fig. S4j, food intake, two-way RM ANOVA, genotype: $F_{1,18} = 5.437$, $p = 0.032$; Fig. S4k, body weight, two-way RM ANOVA, genotype: $F_{1,18} = 7.364$, $p = 0.014$). Importantly, more mutant mice reached the threshold of less than 75% of basal body weight (critical weight loss, as seen in patients with anorexia nervosa) compared to WT mice (Fig. 5o, Kaplan-Meier test, Mantel-Cox and Gehan-Breslow-Wilcoxon post hoc comparison, $p < 0.001$). These results demonstrate that, when VGLUT3$^{T8I/T8I}$ mice have limited access to food in the presence of a running-wheel, they are more likely than WT mice to develop self-starvation behavior.

As above mentioned, VGLUT3$^{T8I/T8I}$ mice exhibit reduced cholinergic transmission in the dorsal striatum (Fig. 3i–n). It was recently reported that donepezil, an acetylcholinesterase inhibitor, can be used to reverse self-starvation phenotype of mice with deficient ACh signaling in the striatum[15]. Mutant mice and WT littermates were treated with donepezil (0.3 mg.kg$^{-1}$ daily) throughout the entire ABA paradigm. Saline and donepezil treatment had no effect on the percentage of control mice reaching the critical threshold of less than 75% of their baseline body weight (Figs. 5p and S3j, k, Kaplan-Meier test, $p = 0.819$, Gehan-Breslow-Wilcoxon test, $p = 0.83$). In contrast, donepezil decreased the percentage of VGLUT3$^{T8I/T8I}$ mice below the 75% threshold of baseline body weight by reversing the decrease in food intake of mutant mice (Figs. 5q and S3j, k Kaplan-Meier test, $p = 0.006$; Gehan-Breslow-Wilcoxon test, $p = 0.004$). Overall, these pharmaco-behavioral experiments with donepezil point to the idea that the cholinergic deficit of VGLUT3$^{T8I/T8I}$ mice drives their vulnerability to self-starvation behavior and suggest that this drug could be repositioned to alleviate EDs in humans.

## Discussion

Substance use disorder is observed in up to 50% of patients with EDs[38–40]. Anorexic patients can be divided in 2 categories: restrictive-type (AN-R) and binge eating/purging (AN-BP) anorexia patients[41]. Interestingly, substance abuse is more frequently reported within bulimia nervosa (BN) patients and AN-BP than among AN-R[38,40]. It has been suggested that this comorbidity could be linked to addictive/impulsive personality trait. However, we lack understanding of neuronal mechanisms that could be common to EDs and SUDs. It has been recently proposed that an imbalance between goal-directed behaviors and habits could lead to loss of control and repetitive self-destructive (compulsive) behaviors such as those seen in EDs and SUDs[3,4,15,42]. ChIs play a complex role in mouse models of habit formation, substance abuse or maladaptive eating[9,10,15,17,30,43]. This complexity is generated by the fact that ChIs signal with ACh and glutamate, and that the two transmitters have opposing effects on the delicate balance between goal-directed behaviors and habits[13,15]. A rare variant of VGLUT3 (VGLUT3-p.T8I) was initially identified in a sample of patients suffering from severe SUDs, who reported African ethnicity[17]. In our cohort of SUDs patients, the p.T8I variant was found in 0.2% in patients of European ancestry, whereas it was present in 11% of patients with African origin. In Africa, it is estimated that 25% of the population use drugs and 3% of them suffer from SUDs[44]. Based on an estimation of over 36 million individuals with SUDs in Africa[44], there could be more than 3 million carriers of p.T8I in Africa alone. In the present study, the p.T8I variant was also identified in a patient suffering from cocaine addiction and BN, suggesting the existence of common genetic markers of vulnerability to both SUDs and EDs.

In this study, we identified 8 patients with severe addiction (Paris SUDs samples) and 1 with bulimia and addiction (Montreal EDs sample) but no AN patient. However, VGLUT3$^{T8I/T8I}$ mice are equally vulnerable to sucrose binge eating test (bulimia-like model) as to self-starvation in the ABA test (anorexia-like model). In humans, ethnicity, associated cultural differences and the effects of migration (thus socio-economic factors) are important risks factor for AN[45] and for SUDs[46]. The p.T8I variant is observed essentially in individuals of African and admixed European-African ancestry; unfortunately, data are lacking in reference panel for the latter. What is known is that the prevalence of AN in Africans is <0.01% compared to 0.7–1% in Europeans and Asians[45,47]. The fact that individuals with Africans ancestry are much less likely to be diagnosed with AN could explain why no anorexic patients were

identified among p.T8I carriers in our samples. Identifying p.T8I carriers with AN of either restrictive (AN-R) or binge-purge (AN-BP) subtype would require exploring very large DNA data bases of ED patients with African ancestors. In Africans, BN and binge eating disorder (BED) patients are more frequent (0.87% and 4.45% respectively) than AN patients. Interestingly we found one BN patient in the EDs sample. Overall, these observations suggest that AN-BP individuals carrying the p.T8I variant could be more frequent than AN-R. However, this remains to be documented in large samples of AN patients from African ancestry.

These observations support the complex interplay of genetic, ethnic and socio-economic factors in the risk for AN and SUDs in humans; they also illustrate the limits of currently available genetic reference panels in humans and of animal models.

The goals of the present study were to establish whether the p.T8I variant was involved in EDs and/or SUDs and then to decipher the molecular and neuronal bases that lead to the two pathologies. Remarkably, VGLUT3$^{T8I/T8I}$ mice exhibited both increased cocaine relapse and maladaptive eating. These findings in our "humanized" mouse model support the idea the p.T8I variant may indeed be causally linked to EDs and SUDs.

At a molecular level, replacing Threonine 8 by an Isoleucine does not quantitatively or qualitatively alter VGLUT3 expression. The co-distribution of VAChT and VGLUT3 in ChIs varicosities has been abundantly documented[12,13]. NNDs analysis between VAChT and VGLUT3 was recently characterized using high-resolution STED microscopy[24]. These investigations suggest that 43% of ChIs synaptic vesicles co-express VAChT and VGLUT3, and therefore have the capacity to co-release ACh and glutamate[24]. We found a similar relative distribution of VGLUT3 and VAChT in striatal cholinergic vesicles from WT mice and VGLUT3$^{T8I/T8I}$ mice. This observation indicates that the p.T8I variation does not severely impair vesicular organization of cholinergic varicosities.

The main function of VGLUT3 is to accumulate glutamate inside synaptic vesicles, therefore allowing exocytotic release of glutamate by ChIs[11,48]. A putative 3D model of VGLUT3 and its N-terminus show that Threonine or Isoleucine in position 8 are positioned away from the central pore of the transporter, and therefore suggest that the variant should not impede glutamate translocation. Indeed, the p.T8I variant does not alter VGLUT3-dependent glutamate vesicular accumulation or release. A second important property of VGLUT3-dependent glutamate is to increase vesicular accumulation and release of ACh[11–13]. This mechanism named vesicular synergy is a powerful presynaptic regulation of cholinergic quantum size since that increases vesicular ACh accumulation by up to 100–200% (present study and[11]). ACh vesicular synergy appears to be partially blunted in ChIs from VGLUT3$^{T8I/T8I}$ mice, without affecting glutamate accumulation. A current model to explain vesicular synergy is based on the glutamate-dependent acidification of VGLUT-positive cholinergic or monoaminergic synaptic vesicles[11,12,49,50]. Since the p.T8I mutation partially reduces vesicular synergy without affecting glutamate accumulation implies that this vesicular regulatory mechanism could rely on both glutamate-dependent acidification of synaptic vesicles and on another yet unidentified mechanism. The fact that the mutation of Isoleucine at position 8 in the N-terminus portion of VGLUT3 impairs vesicular synergy points to the idea that this region of VGLUT3 is involved in vesicular synergy and may interact directly or indirectly with a partner that activates VAChT-dependent ACh vesicular accumulation. Therefore, molecular mechanisms underlying vesicular synergy are likely not fully understood and more complex than currently thought. Alternative explanations for vesicular synergy should be explored in future investigations to clarify this point. This is important since our present study reveals that a decreased vesicular synergy could have dramatic consequences (Fig. S5).

In line with the partial loss of vesicular synergy, we observed a significant decrease of ACh efflux in the DMS of VGLUT3$^{T8I/T8I}$ mice. Reducing ACh release by specific deletion of VAChT in ChIs (VAChTcKO mice) was previously shown to decrease DA efflux in the DMS and the NAc, but not in the DLS[15,17]. Strikingly, VGLUT3$^{T8I/T8I}$ mice show the same pattern of changes in DA release as VAChTcKO mice, i.e. reduction in the DMS versus no change in the DLS. Therefore, VGLUT3$^{T8I/T8I}$ mice have an asymmetric pattern of DA signaling in the dorsal striatum, promoting DLS/putamen activity and habit formation (see the hypothetical model proposed in Fig. S5 and[15]). In addition, VGLUT3$^{T8I/T8I}$ mice are more vulnerable to cue-induced relapse of cocaine-seeking, binge-like sucrose overconsumption and self-starvation.

Interestingly, VGLUT3$^{T8I/T8I}$ mice and WT mice show similar vulnerability to food addiction and comparable levels of anhedonia and anxiety. Mutant mice display normal behavior (including eating behavior) in their everyday environment. Therefore, increased vulnerability to cocaine reinstatement and maladaptive eating of VGLUT3$^{T8I/T8I}$ mice might be related to concomitant adverse conditions or stress and/or to early life adversity (Fig. S5).

An important question is to determine whether these findings are limited to carriers of the p.T8I allele or if they can be generalized to other groups of patients carrying different variations. Subtypes of alpha subunit of the nicotinic receptor (nAChR) are expressed on DA fibers in the dorsal striatum where they powerfully regulate DA efflux[20,29,51]. Interestingly, several single nucleotide polymorphisms in the α5nAChR are associated with nicotine addiction, cocaine use disorders as well as potentially with EDs[52–54]. This polymorphism has an allelic frequency of 42% in populations with European ancesters[55]. Rat with a knockout of the α5nAChR exhibit increased vulnerability to cue-induced cocaine relapse[53], a feature also observed with VGLUT3$^{T8I/T8I}$ mice. Similarly, transgenic rats expressing the rs16969968 α5nAChR variant consume more alcohol and show increased cue-induced relapse to alcohol and food seeking after a phase of abstinence[52]. These data suggest that a loss of function of the α5nAChR subunit could be associated with SUDs and maladaptive eating. It is not yet established in these rodent models whether polymorphisms of the α5nAChR unbalance DA transmission in DMS vs DLS compartments and increase habit formation. However, these previous findings suggest that the model depicted in our study could be generalized to a large portion of the human population.

Beyond the fact that p.T8I blunts vesicular synergy and ACh striatal release, we propose that a common mechanism leading to several severe psychiatric disorders could be the imbalanced transmission of DA in DMS/caudate versus DLS/putamen (Fig. S5). This asymmetric striatal DA transmission might be an overlapping feature of EDs and SUDs, but also more widely to psychiatric disorders with a compulsive component. This model provides a framework to understand and alleviates several major psychiatric disorders.

Anorexia nervosa has one of the highest mortality rate among psychiatric disorders, either by suicide or somatic complications, with an average of 1% of death every year[56]. However, we critically lack specific treatments to efficiently improve anorexic patients. Even though this may be controversial, acetylcholinesterase (AChE) inhibitors have been proposed as a potential treatment for obsessive-compulsive disorders, for pathological repetitive behaviors and for eating compulsions in Alzheimer's disease[57–62]. We previously reported that donepezil was able to rectify self-starvation observed with VAChTcKO mice in the ABA model[15]. Data from the literature combined with our present findings suggest that donepezil could be repositioned to alleviate anorexia nervosa symptoms. Procholinergic treatments might also be relevant in other EDs or in SUDs patients. Interestingly, we collected recent and unpublished observations suggesting that donepezil demonstrates clinical efficacy in several case reports with patients suffering from anorexia nervosa. However,

donepezil can cause hypertension, cramps, nausea and diarrhea and should therefore be used with caution to treat a population of young and undernourished patients. In particular, low doses of donepezil should be favored in potential future clinical trials.

Altogether, our results show that VGLUT3[T8I/T8I] mice recapitulate pathological phenotypes reported in patients with EDs or SUDs and provide a unique model to understand common molecular basis of these life disrupting pathologies. Increasing ACh transmission with a treatment like donepezil might help restore balanced striatal DA transmission and improve patients with EDs, but also likely other compulsive disorders.

# Methods

## Human genetic analysis and clinical observations

We performed candidate-gene association study driven by preclinical evidence. We used both a within-cases and a case-control design, aimed at investigating the genetic associations between *SLC17A8* (the gene encoding VGLUT3) variants and cocaine-related phenotypes.

**Patient recruitment.** We investigated the phenotypic correlates of *SLC17A8* variants in two independent samples of genotyped patients and healthy controls who each underwent extensive characterization for addictive and mental disorders. These samples were chosen since their participants were recruited for disorders showing a strong compulsive component, which likely represents a transdiagnostic endophenotype underlying the vulnerability to several mental disorders[63,64].

The samples were:

- *Sample #1 eating disorders* (hereafter named EDs sample), recruited by the Douglas hospital eating disorders program, including healthy controls (n = 71) and 270 out- and inpatients with diagnosis for anorexia nervosa (n = 74) bulimia nervosa (n = 116), eating disorders not otherwise specified (EDNOS, n = 80), mean age=25 ± 0.4 year-old[65]; (details provided in Supplementary Methods and Supplementary Table 6). All were women.

- *Sample #2 substance use disorders* (hereafter named SUDs sample), recruited in specialized SUDs outpatient clinics 524 outpatients (78% men) seeking treatment for SUDs, mean age=43 ± 9-year-old, genotyped by DNA array for 321 rare and frequent variants of *SLC17A6* (VGLUT2), *SLC17A7* (VGLUT1) and *SLC17A8* (VGLUT3) including p.T8I (details provided in Supplementary Methods and Supplementary Table 7).

In both samples, extensive phenotyping was performed by trained researchers/clinicians during a single face-to-face interview based on structured assessment instruments. EDs and/or SUDs were ascertained, based on DSM-IV-TR criteria (such that the SUDs diagnoses of interest corresponded to substance dependence[66], using the Eating Disorders Examination, a semi-structured interview assessing the presence and severity of core EDs symptoms[67]; and the Structured Clinical Interview for DSM-IV (SCID-IV), respectively. Additionally for the SUDs sample, the Scale for Assessment of Positive Symptoms-Cocaine-Induced Psychosis (SAPS-CIP) was used to measure motor and psychotic complications of cocaine use[68,69].

Both protocols were preregistered and approved by local ethics committees (CPP Ile-de-france IV & VI), in accordance with the Helsinki declaration as revised in 1989 (Supplementary Material and Methods). All participants were asked for written informed consent for initial and follow-up genetic analyses.

**Biological sampling and genotyping.** The DNA from healthy controls and patient was extracted from whole blood (EDs sample) or peripheral blood cells (SUDs sample). After amplification by polymerase chain reaction, DNA underwent thorough quality control and was conserved according to the gold standard specifics from each country. After genotyping, both markers and samples with missing genotypes >2% were filtered out. Ancestry was determined by self-report in EDs

sample and by genotyping in SUDs sample, according to standard procedures. SUDs sample was further controlled for the absence of relatedness down to 3rd degree and respect of Hardy-Weinberg Equilibrium.

**Inclusion and ethics.** All procedures performed in studies involving human participants or animals were conducted in accordance with the ethical standards of Declaration of Helsinki and in agreement with local ethic committees. All participants provided written informed consent to participate in the study.

## Animals

Animal studies were performed in accordance with the European Communities Council Directive (86/809/EEC) regarding the care and use of animals for experimental procedures in compliance with the *Ministère de l'Agriculture et de la Forêt, Service Vétérinaire de la Santé et de la Protection Animale* (authorization #13713-2018021516201278). Animal care, handling and all experiments were performed according to the guidelines of the Canadian Council on Animal Care (http://ccac.ca/en_/standards/guidelines) and approved by the Facility Animal Care Committee of the Douglas Research Center (protocols 2008-5643 and 2014-7479) and *Comitè Ètic d'Experimentació Animal-Parc de Recerca Biomèdica de Barcelona* (CEEA-PRBB, Protocol Number: RML-16-0048-P1) and *Generalitat de Catalunya* (Protocol Number: DAAM-9687). *Animal-Parc de Recerca Biomèdica de Barcelona* facilities have Animal Welfare Assurance (#A5388-01, Institutional Animal Care and Use Committee approval date 05/08/2009) granted by the Office of Laboratory Animal Welfare (OLAW) of the US National Institutes of Health. In the present study we used 235 male mice (121 wildtype (WT) mice and 114 VGLUT3[T8I/T8I] mice). Male mice were used since we previously reported that hypocholinergic phenotypes are similar in males and females[15]. However, investigations are currently undergoing to compare males and females VGLUT3[T8I/T8I] mice. For each biochemical or behavioral mouse study, sample size determination was based on previous investigations[15,70]. All efforts were made to minimize the number of animals used during our studies and to ensure their well-being. Animals were housed in a temperature-controlled room (21 ± 2 °C, 30–40% humidity) with *ad libitum* access to water and food under a 12 h light/dark cycle. All biochemical and behavioral tests were conducted with 2–8 months old littermate mice.

## Construction of the mutant, genotyping and breeding of WT mice and VGLUT3[T8I/T8I] mice

A mouse line expressing the p.T8I point mutation in the *Slc17a8* gene was generated at Phenomin–Institut Clinique de la Souris (Illkirch, France; http://www.phenomin.fr/) and named VGLUT3[T8I/T8I]. A point mutation was introduced in exon 1 of the mouse *Slc17a8* gene, leading to the ACC codon (coding for a threonine) in exchange for the ATC codon (coding for an isoleucine) (Fig. 1e–g).

Mice were genotyped by PCR analysis with the following PCR primers:

Mf: 5'-GAATTTCAAGTGTTCCTCCAGGGCAA-3'
Mr: 5'-TCCCACGGCATTCTTCACTCCTT-3'
Ef: 5'-TGTTAGGAATATCACTCACTGCTGGTGCTA-3'
Er: 5'-CGGTCTTGGAATTTTCCCACTGCTA-3'

PCR amplification yielded bands of 392 and 471 bp for WT and the mutated allele respectively (Fig. 1f).

Mice were obtained by crossing heterozygous male and female mice (C57BL/6 N genetic background). Animals were randomly allocated to experimental groups. Whenever possible, investigators were blinded to genotypes during experimental procedures.

## Mutagenesis and construction of plasmids

To introduce a point mutation in the WT alleles, we used the Quik-Change II XL Site-Directed Mutagenesis kit (Stratagene) and a set of

complementary primers as described previously[70]. All clones were sequenced in both directions, and the plasmids were purified using the Plasmid Maxi kit (Qiagen) before use.

## Hippocampal neuronal culture

For immunofluorescence experiments, hippocampal cell cultures were prepared from newborn P0 (the day of birth) C57BL/6N wildtype mice (WT) or VGLUT3[T8I/T8I] pups of either sex as described previously[70]. Neurons were seeded at a density of 1000 neurons per well of 12-wells dish with coverslips previously coated with poly-L-lysine solution. After 7 days in culture, neurons were transfected with linearized plasmid pcDNA3-VGLUT3-IRES-GFP or pcDNA3-VGLUT3-p.T8I-IRES-GFP with Lipofectamine 2000 (Thermo Fisher Scientific; 1 µg of DNA for 1 µl of lipofectamine). At 9 days in vitro (DIV), neurons were fixed with 5 min exposition to paraformaldehyde 4%, washed with PBS 1X and stored at 4 °C until immunofluorescence experiments were performed.

## Immunofluorescence

Immunofluorescence experiments on hippocampal primary neurons and brain slices were performed as described[70,71]. In brief, immunocytochemistry protocols were performed on hippocampal neurons after transfection protocol, fixation with 5 min exposition to paraformaldehyde 4%, washes with PBS 1X and storage at 4 °C. Immunohistochemistry protocols were performed on dissected brain from anesthetized and intracardially perfused mice with 0.1 M phosphate-buffer containing 2% paraformaldehyde. Brains were post-fixed by immersion in the same fixative for 48 h. Coronal sections (50 µm) were cut using a vibratome and stored at 4 °C. Coverslips seedings cells or free-floating brain sections were incubated with anti-human VGLUT3 rabbit polyclonal antiserum (1:1000)[72], anti-VGLUT3 rabbit polyclonal antiserum (1:2000, Synaptic Systems), anti-rodent VGLUT1 rabbit polyclonal antiserum (1:2000)[73], anti-rodent MAP2 mouse monoclonal antiserum (1:1000, Sigma), anti-rodent bassoon mouse monoclonal antiserum (1:2000, Abcam), or PSD-95 mouse monoclonal antiserum (1:2000, Abcam). Immunolabeling was detected with anti-rabbit or anti-mouse secondary antisera coupled to Alexa Fluor 555 or Alexa Fluor 488 (1:2000, Invitrogen). Nuclei were labeled using DAPI (1:5000, Sigma). Confocal microscopy imaging was performed using a Leica TCS SP5 (Leica Microsystems) at the Imaging facility of the IBPS. Images were acquired in the dorsolateral striatum (DLS) using a hybrid detector (HyD) and a x63 oil immersion objective, NA 1.4 (with a 5x zoom).

## Immunoautoradiographic labeling of VGLUT3

Immunoautoradiography experiments were performed on fresh frozen mouse brain sections (10 µm) as described previously[17,70,74]. In brief, brain series were incubated with VGLUT3 rabbit polyclonal antiserum (1:20,000, Synaptic Systems) and then with anti-rabbit [125I]-IgG (PerkinElmer). The sections were then washed in PBS, rapidly rinsed in water, dried, and exposed to x-ray films (Biomax MR, Kodak) for 5 days. Standard radioactive microscales were exposed to each film to ensure that labeling densities were in the linear range. Densitometry measurements were performed with MCID analysis software on six to eight serial sections for each region (striatum, hippocampus and raphe nuclei) per mouse (four mice per genotype).

## Stimulated Emission Depletion (STED)

Preparation of synaptic vesicles were visualized with STED microscopy as previously reported[24]. In brief, synaptic vesicles were adsorbed on glass slides and incubated with a mixture of primary anti-VGLUT3 and anti-VAChT antiserums (dilution 1:5000 (VAChT) and 1:2000 (VGLUT3). Synaptic vesicles incubated with secondary antibodies coupled to Alexa-594 and Abberior-Star 635 P fluorophores (Abberior GmbH, Göttingen Germany; dilution: 1: 1000), fixed in paraformaldehyde (PFA 2%) and

mounted on slides with ProLong Diamond Antifade Mountant (P36961, Thermo Fisher Scientific). Images were acquired with a STED super resolution microscope (LEICA SP8 – STED 3X, equipped with a 775 nm depletion laser; Leica Microsystems).

## Homology modeling of VGLUT3-p.T8I

Based on the x-ray crystal structure of the E. coli D-galactonate:proton symporter (DgoT)[25] a member of the solute carrier 17 (SLC17) family and distant ortholog of vesicular glutamate transporter, a putative 3D structure of VGLUT3 was established as previously described[70,75–77]. In brief, the mutant model was minimized using the Adopted Basis Newton-Raphson (NR) algorithm, with a maximum step of 500 and a "generalized Born with Implicit Membrane" as an implicit solvent model. A 1-palmitoyl-2-oleoyl-phosphatidylcholine membrane of 100 × 80 Å was added. Proteins were solvated, and ions were added using the solvation and ionization package from Biovia Discovery studio (Dassault Systemes). The system was then equilibrated using a short Nanoscale Molecular Dynamic software program (NAMD) of 1 ns.

## Neuronal microculture and electrophysiological recording of autapses

Hippocampi were harvested at postnatal day 0 (P0) to P1 from VGLUT1 knock-out mice (VGLUT1[−/−] mice) of either sex[26]. Neurons were plated on island cultures at a density of 2000–3000 neurons per 35 mm dish. Recordings were performed from 14 to 18 days in vitro (DIV). VGLUT1[−/−] autaptic neurons were infected with either VGLUT3 or VGLUT3-p.T8I lentiviral vectors (10 and 40 ng of p24 per well, respectively). The standard extracellular solution contained the following (in mM): 140 NaCl, 2.4 KCl, 10 HEPES, 10 glucose, 4 MgCl2, and 2 CaCl2, pH 7.3. The internal solution contained the following (in mM): 135 KCl, 18 HEPES, 1 EGTA, 4.6 MgCl2, 4 ATP, 0.3 GTP, 15 creatine phosphate, and 20 U/ml phosphocreatine kinase. EPSCs were evoked by 2 ms of depolarization at 0 mV, resulting in an unclamped action potential. Vesicular release probability was computed by dividing the EPSC charge by the RRP charge. Current traces were analyzed using Axograph X, Excel (Microsoft), and Prism (GraphPad Software). The mEPSCs were detected with a template function (Axograph; template: rise, 0.5 ms; decay, 3 ms; criteria range: rise, 0.15–1.5 m; decay, 0.5–5 ms).

## Vesicular [3H]ACh or [3H]Glutamate vesicular uptake in mouse striatal synaptic vesicles

Synaptic vesicle isolation from mouse striatum and uptake assays of [3H]ACh or [3H]Glutamate were performed as described previously[11,78]. In brief, transport reactions were initiated by adding 10 µl of striatal synaptic vesicles (50 µg of protein) to 90 µl of uptake buffer in the presence or absence of ATP (2 mM, Sigma-Aldrich) and [3H]Glutamate (1.2 µci, 400 nM, PerkinElmer), [3H]ACh (2 µci, 30 nM, PerkinElmer) with or without 50 µM vesamicol (Sigma-Aldrich) or L-glutamate (10 µM, Sigma-Aldrich). After 10 min at 37 °C, vesicular uptake was stopped by dilution in 3 ml of ice-cold transport buffer, rapid filtration through mixed cellulose esters filters (MF; Millipore), and three washes with 3 ml of ice-cold 0.15 M KCl. Radioactivity retained on the filters was measured by scintillation counting. Each determination was performed in triplicate, and independent experiment were performed three times using two different synaptic vesicles preparations.

## In vivo fiber photometry of ACh

To assess in vivo ACh dynamics in basal conditions, real-time fluorescence intensity was recorded using the m4-based genetically encoded GACh4.3 ACh biosensor[27]. WT mice or VGLUT3[T8I/T8I] mice were injected with AAV1 expressing the ACh sensors under the control of a synaptophysin promoter to specifically target neurons (GACh4.3, 400nL, 3.57*10[14]M)[27]. Mice were implanted with optical fibers (200 µm core NA 0.37, Doric Lenses) for fiber photometry recordings and fluorescence measurements were recorded using Doric Lenses 1-site 2-color Fiber

Photometry System. The stereotaxic coordinated used for both virus injection and fiber implantation were as follow: AP, −1.00 mm: ML, +/−1.2 mm; DV, −3.00 mm. The fiber photometry console was connected to the LED driver to control the two connectorized LEDs in Lock-in mode (405 nm LED modulated at 333.786 Hz and 465 nm LED modulated at 220.537 Hz). Each LED was connected to its respective port on the Mini Cube through an optic patchcord. Light stimulation and recorded fluorescence were transmitted through an optic fiber (400 μm core NA 0.39, Thorlabs) connected both to the animal's implanted optic fiber via a zirconia sleeve and to the sample port on the Mini Cube. Finally, the photoreceiver converting recorded light to electrical signals (AC Low setting, New Focus 2151 Visible Femtowatt Photoreceiver, New Focus, San Jose, CA, USA) was connected to the Mini Cube through an optic patchcord (600 μm core NA 0.48 FC-FC, Doric Lenses) fitted on a fiber optic adapter (Doric Lenses) and to the fiber photometry console. Signal was acquired through Doric Neuroscience Studio software (version 5.2.2.5) with a sampling rate of 12.0 kS/s (kilosamples per second) and a low-pass filter with a cutoff frequency of 12.0 Hz. All fiber photometry data were analyzed on R software. Data were downsampled and the mean value of "auto-fluorescence" (signal acquired after each recording with the same parameters, but the optic fiber not connected to the mouse) was subtracted from the signal. An exponential fit of the signal was subtracted to it before adding an offset equal to the mean of the signal before detrending to account for the slow decay of the signal due to photobleaching during the recording. The control channel (405 nm) was subtracted to the GACh signal (465 nm) and resulting $\Delta F/F$ was z-scored.

To detect non-event locked cholinergic events, we used a peak detection routine adapted from Holly et al.[79]. We applied a rolling median with a 10 s window to our signal and filtered out high amplitude events (amplitude greater than twice the median absolute deviation (MAD) above the rolling median). We then calculated the median of the filtered trace. Peaks with local maxima over 3 MADS above this median were detected and used for analyses of amplitude and frequency.

### In vivo voltammetry of DA release
In vivo voltammetry was performed as previously described[15,17]. In brief, mice were anaesthetized with chloral hydrate (400 mg.kg⁻¹, i.p.) and voltametric electrodes (Aldrich, Milwaukee, WI, USA) were implanted into the dorsomedial (DMS, stereotactic coordinates in mm relative to bregma: AP, +1.1; ML, ±1; relative to the dura, DV, −2.6) or dorsolateral striatum (DLS: AP, +1.1; ML, ±2; DV, −2.6). Voltammetric electrodes consisted of one 30-μm diameter carbon fiber coated with Nafion (Aldrich). Electrochemical measurements were performed using a high-speed chronoamperometric apparatus (Quanteon, Lexington, KY, USA) as previously described[80]. In brief, DA release was evoked by local injection of 200–300 nl of KCl (120 mM). The local depolarization by a KCl puff was favored over receptor piloted DA release (triggered by nicotinic agonist or by mGLUR antagonist[15,17]) since it was faster and necessitated a smaller injection volume (200–300 nl) than infusions of pharmacological compounds. Results are presented as the mean ± SEM of the difference in maximal DA variation after KCl ejection. In vivo voltammetry data were collected with FAST (Quanteon System 3, Quanteon, Lexington, Kentucky, USA). Data were then converted to excel analyzed with Prism. The differences in DA release between the different groups of mice were assessed as a comparison with the differences in maximal variation for each group.

### Behavioral experiments
**Elevated Plus Maze (EPM).** The EPM was used to measure unconditioned anxiety-like behavior[16]. The EPM, which consisted of two open arms, two enclosed arms, and a central platform elevated 38.5 cm above the ground, was placed into 10 lux ambient light. After being allowed 1 h of habituation in the testing room, the animals were placed in the central area, facing one of the open arms and were tested for 360 s. The number of entries in close or open arms, the time-spent in open arms and the number of transitions from close to open arms were measured (Viewpoint, France).

**Elevated O-maze (O-Maze).** The O-maze was used to measure unconditioned anxiety-like behavior[81]. The O-maze test was performed in a white elevated O-shaped arena (7 cm wide arena on a 47 cm diameter circle, 52 cm height). Two opposed quarters have walls (closed arms) and the two other ones have not (open arms, anxiogenic zone). Light intensity was 10lux in the open arms. Mice were introduced into one of the closed arms and allowed to freely explore the arena for 600 s. The number of entries in closed or open arms and the time-spent in open arms were measured.

**Marble burying.** Marble burying test was performed as previously described with minor modification[16]. In brief, the floor of clear Plexiglas box (14 × 32 × 13 cm) was covered with a 5 cm layer of sawdust bedding. Fifteen glass marbles (13 mm diameter) were evenly spaced in 5 rows of 3. Mice were individually placed in the cage for 15 min. The number of marbles more than two-thirds covered with sawdust was recorded every minute.

**Basal locomotor activity.** Spontaneous locomotor activity was assessed as previously described[70] by placing mice in activity boxes (20 × 10 × 15 cm³) for 5 h of habituation followed by 24 h of "night/day recording". Horizontal and vertical activities were measured by photocell beams located across the long axis, 15 mm (horizontal activity) and 30 mm (vertical activity) above the floor. Each box was connected by an interface to a computer (Imetronic, France). Locomotor activity was measured in 1 h intervals during habituation and from 7:00 a.m to 7:00 p.m. Food and jellified water were provided ad libitum.

**Operant cocaine self-administration, extinction and reinstatement.** VGLUT3^T8I/T8I knock-in mice and their WT littermates were trained for cocaine self-administration (0.5 mg.kg⁻¹ per infusion) during continuous reinforcement using a fixed-ratio 1 (FR1, one injection for each nosepoke) or 3 (FR3, one injection for 3 nosepokes) then motivation for cocaine (progressive ratio, PR), extinction and cue-induced reinstatement were evaluated[17].

**Operant sucrose self-administration and devaluation training.** Operant sucrose self-administration was performed as previously described[15]. In brief, initial nosepoke training consisted of 16 consecutive days of continuous reinforcement FR1 procedure during 60 min without cutoff. A sensory-specific satiety procedure was used during 2 days for outcome devaluation[82]. On each day, mice were given for 1 h ad libitum access to home chow or sucrose pellets reinforcer (devalued condition). The amount of food consumed during the ad libitum session was recorded, and mice that did not consume a minimum of 0.4 g of each food were not included in the analyses. Immediately after the ad libitum feeding session, mice were given a 5-min test in extinction in the operant chambers, but no pellet was delivered. The order of the valued and devalued condition tests (day 1 or day 2) was counterbalanced across animals, and the number of active and inactive nosepokes for each condition was recorded.

### Food addiction
**Operant training maintained by palatable food.** The beginning of each operant training session was signaled by turning on a house light placed on the chamber ceiling during the first 3 s. Daily operant training sessions maintained by chocolate-flavored pellets lasted 1 h. The operant training sessions were composed of two pellet periods

(25 min) separated by a pellet-free period (10 min). Pellets were delivered contingently after an active response paired with a stimulus light (cue light) during the pellet periods. A time-out period of 10 s was established after each pellet delivery, where the cue light was off, and no reinforcer was provided after responding on the active lever. Responses on the active lever and all the responses performed during the time-out period were recorded. During the pellet-free period, no pellet was delivered, and this period was signaled by the illumination of the entire operant chamber. In the operant conditioning sessions, mice were seven days under fixed-ratio 1 (FR1) schedule of reinforcement (one lever-press resulted in one pellet delivery) followed by an increased FR to 5 (FR5) (five lever-presses resulted for one pellet delivery) for the remaining 118 sessions. Three behavioral tests were used to evaluate the food addiction-like criteria recently described[35,83,84]. These three criteria summarized the hallmarks of addiction based on DSM-IV, specified in DSM-5, and now included in the food addiction diagnosis through the YFAS 2.0. Food addiction criteria were evaluated during the early training period (sessions 1–18 of FR5) and also during the late training period (sessions 95–118 of FR5).

**Persistence to response.** Non-reinforced active responses during the pellet-free period (10 min), when the box was illuminated and signaling the unavailability of pellet delivery, were measured as persistence of food-seeking behavior. On the three consecutive days before the progressive ratio, mice were scored.

**Motivation.** The progressive ratio schedule of reinforcement was used to evaluate the motivation for the chocolate-flavored pellets. The response required to earn one single pellet escalated according to the following series: 1, 5, 12, 21, 33, 51, 75, 90, 120, 155, 180, 225, 260, 300, 350, 410, 465, 540, 630, 730, 850, 1000, 1200, 1500, 1800, 2100, 2400, 2700, 3000, 3400, 3800, 4200, 4600, 5000, and 5500. The maximal number of responses that the animal performs to obtain one pellet was the last event completed, referred to as the breaking point. The maximum duration of the progressive ratio session was 5 h or until mice did not respond on any lever within 1 h.

**Compulsivity.** Total number of shocks in the session of shock test (50 min) performed after the PR test, when each pellet delivered was associated with punishment, were used to evaluate compulsivity-like behavior, previously described as resistance to punishment[35]. Mice were placed in a self-administration chamber without the metal sheet with holes and consequently, with the grid floor exposed (contextual cue). In this shock-session, mice were under an FR5 schedule of reinforcement for 50 min with two scheduled changes: at the fourth active lever-response, mice received only an electric footshock (0.18 mA, 2 s) without pellet delivery, and at the fifth active lever-response, mice received another electric footshock with a chocolate-flavored pellet paired with the cue light. The schedule was reinitiated after 10 s pellet delivery (time-out period) and after the fourth response if mice did not perform the fifth response within 1 min.

**Attribution of the three addiction-like criteria.** After performing the three behavioral tests to measure food addiction-like behavior, mice were categorized as addicted or non-addicted animals depending on the number of positive criteria they had achieved. An animal was considered positive for an addiction-like criterion when the score of the specific behavioral test was above the 75th percentile of the normal distribution of the WT control group. Mice that achieved 2 or 3 addiction-like criteria were considered addicted animals, and mice that achieved 0 or 1 addiction-like criteria were considered non-addicted animals.

**Binge-like sucrose overconsumption model.** Sucrose binge-like phenotype in mice were evaluated according to Yasoshima et al.[37]

and as previously reported[15]. In brief, mice were housed individually in Plexiglas cages. For habituation, water and food were available *ad libitum* for 2 days. Mice were then food deprived for 20-h (12 p.m–8 a.m.) with free access to water. During 4 h per day (8 a.m.–12 p.m.), mice had simultaneously access to a highly concentrated sucrose solution (20%, Sigma) and water with the two-bottle method and food. To control for side preference, the left/right position of the sucrose solution and water was alternated daily. Consumption of the sucrose solution during the first hour (8 a.m.–9 a.m.) and the total 4 h (8 a.m.–12 p.m.) were recorded daily. Consumptions of water and food were recorded daily after 4 h access. Daily body weight (BW) was also measured before sucrose and food access. Another cohort of mice received saccharine (5 mM) instead of sucrose throughout the same regimen.

**Sucrose preference.** Mice were given 24-h concurrent access to two graduated plastic bottles for 3 days in their home cages[15]. One of these bottles contained tap water, whereas the other one contained 20% sucrose solution. Bottles were weighed daily, with the position of the bottles (left/right) alternated to control for side preference. The first day was used as a habituation period. The volumes of sucrose solution and water consumed on the second and third days were averaged to determine sucrose, water and total fluid intake, and preference for sucrose over water (sucrose intake/total fluid intake).

**Activity-based anorexia (ABA) model.** The ABA model was performed as described by Klenotich and Dulawa[36] and as previously reported[15]. In brief, for habituation, all mice were individually housed in cages with running wheels for 7 days with unrestricted access to food, water and running wheel. After the adaptation period, mice were maintained in the same cages for 8 additional days. Access to food was progressively restricted, from 8 h (day 1) to 2 h (day 8) per day. Body weight and food intake were measured daily before and after food access, respectively. Days until mice reached 75% of baseline BW provided a measure of survival. For pharmacological experiment, mice received intraperitoneal injection of donepezil (0.3 mg.kg$^{-1}$, diluted in NaCl 0.9%) or NaCl 0.9%. Mice were treated daily 30 min before the start of food access, during both baseline and food restriction phases.

## Statistics

**Statistical analyses of clinical data.** A case-control study was performed to compare allelic frequencies of p.T8I in pooled clinical samples to the gnomAD multi-ancestry reference panel (806,886 individuals genotyped by whole-genome array or sequencing (https://gnomad.broadinstitute.org).

In the SUDs sample, we investigated whether the *SLC17A8* rs45610843-T variant (p.T8I) was specifically associated with severe SUDs phenotypes using a within-cases study design. Phenotypes of interest were the presence of lifetime cocaine, alcohol, opioid, benzodiazepine or cannabis use disorder, the total score on the SAPS-CIP[81,82] and scores for each of four subscales: hallucinations, delusions, physical symptoms before cocaine use and motor disturbances. Patients with the p.T8I variation (n = 8) were compared to patients carrying missense mutations of *SLC17A8* (n = 7) and to healthy controls[64]. Additionally, we compared patients with or without frequent allelic mutations of *SLC17A8* (170 SNPs, Supplementary Table 9) amongst patients of European ancestry only. Group differences were tested by Chi-squared/Fisher's exact or Kruskal-Wallis tests, when appropriate. Analyses regarding rare mutants were corrected for multiple testing in pairwise comparisons only, due to small sub-groups sizes. For SNPs analyzed in European individuals, Bonferroni correction was applied to p-values and regression analysis was further adjusted on the 10 first principal ancestry components. R version 4.2.0 was used for analyses (session summary available as a Supplementary File).

**Statistical analyses of animal data.** All statistical comparisons were performed with two-sided tests with Prism 9 (GraphPad Software). Normality and equality of variances were checked for all datasets to select appropriate statistical test for the relevant experimental design. Student t-test or Mann-Whitney test, Kruskal-Wallis test, Kaplan-Meier test, one-way and (repeated-measures, RM) two-way analysis of variance (ANOVA) were used when appropriate. Bonferroni's test, Dunn's test, the method of contrasts, log-rank Mantel-Cox and Gehan-Breslow-Wilcoxon tests or Tukey's test for post hoc analysis were performed when required. Measurements were taken from distinct samples.

## Reporting summary

Further information on research design is available in the Nature Portfolio Reporting Summary linked to this article.

## Data availability

Clinical data and human genetic individual-level genotypes cannot be made available for ethical reasons. Summary statistics and data underlying the figures are available upon request to Romain Icick (romain.icick@aphp.fr). Raw data from all molecular, cellular or mouse investigations are available without restrictions upon request to Salah El Mestikawy (salah.elmestikawy@mcgill.ca). All unique biological materials (knock-out mice or antiserums) used are readily available upon request to Salah El Mestikawy. Matlab custom code for acetylcholine fiber photometry are available upon request to Alexandre Mourot (alexandre.mourot@espci.psl.eu). Source data are provided with this paper.

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

## Acknowledgements

The authors are grateful to the IBPS Animal facility for breeding and care, and the IBPS-NPS Phenotypic core facility where part of the behavioral experiments was performed. **Funding**: SEM: HBHL Neuro Commercialization Grants – Ignite, Natural Sciences and Engineering Research Council Discovery Grants (RGPIN/386431-2012 and RGPIN/04682-2017), SEM, SJ, FV, RM: ERA-NET NEURON Joint Transnational Call for "European Research Projects on Mental Disorders" and research projects on synaptic dysfunction in disorders of the central nervous system (JTC 2013 and 2017, CNRS, INSERM and Sorbonne Université. SD: This project is partly supported by UNAFAM and the Bouygues group with the scientific expertise provided by the FRC's scientific council (AO 2019). SEM, SJ, FV: Agence Nationale de la Recherche (ANR-13-SAMA-0005-01). NP, SD: 80 Prime (CNRS). JJ: Fondation de la recherche Médicale (FRM) for the PhD fellowship, and the Labex Memolife for the 4th year of PhD fellowship. AM: Labex Memolife starting package, and FRM Equipe (FRM EQU201903007961)

## Author contributions

MF, LP, PHD, RI, AH, SJ, FV performed and analyzed genetic and clinical characterizations. MF, CA, CM, SR, NMG, FH, AG, SD performed and analyzed behavioral experiments. EMG performed and analyzed cocaine self-administration and food addiction experiments. JZ, CR performed and analyzed electrophysiological recordings. MF, NMG, OP, MD, VB, VF performed and analyzed anatomical experiments. MF, OP, SD, NP performed and analyzed vesicular uptake. MF performed and analyzed DA voltammetry. YL contributed new reagent and analytical tools. MF, CA, JJ, SM performed and analyzed ACh fiber photometry. MF, EMG, RI, CA, JJ, CR, RM, SJ, FV, AM, NP, SD and SEM designed the study and wrote the manuscript. All authors were involved in revising the manuscript for intellectual content. All authors red and approved the final manuscript.

## Competing interests

The authors declare no competing interests.

## Additional information

[1]Douglas Mental Health University Institute, Department of Psychiatry, McGill University, Montreal, QC H4H 1R3, Canada. [2]Laboratory of Neuropharmacology-Neurophar, Department of Medicine and Life Sciences, Universitat Pompeu Fabra (UPF), Barcelona, Spain. [3]Hospital del Mar Medical Research Institute (IMIM), Barcelona, Spain. [4]Departament de Psicobiologia i Metodologia de les Ciències de la Salut, Universitat Autònoma de Barcelona, Cerdanyola del Vallès, 08193 Barcelona, Spain. [5]Département de Psychiatrie et de Médecine Addictologique, DMU Neurosciences, APHP.Nord, Assistance Publique - Hôpitaux de Paris (AP-HP), Paris F-75010, France. [6]INSERM U1144, "Therapeutic optimization in neuropsychopharmacology", Paris F-75006, France. [7]Université Paris Cité, Inserm UMR-S1144, Paris F-75006, France. [8]Neurobiologie Intégrative des Systèmes Cholinergiques, Département de Neurosciences, Institut Pasteur, Paris F-75015, France. [9]Sorbonne Université, INSERM, CNRS, Neuroscience Paris Seine – Institut de Biologie Paris Seine (NPS – IBPS), 75005 Paris, France. [10]Brain Plasticity Unit, CNRS UMR 8249, ESPCI Paris, PSL Research University, 75005 Paris, France. [11]Neurocure NWFZ, Charite Universitaetsmedizin, Institut für Neurophysiologie, Charitéplatz 1, 10117 Berlin, Germany. [12]Fondation FondaMental, Créteil, France. [13]Université Paris Est Créteil, INSERM, IMRB, Translational Neuropsychiatry, F-94010 Créteil, France. [14]State Key Laboratory of Membrane Biology, Peking University School of Life Sciences, Beijing, China. [15]GHU Paris

Psychiatrie et Neurosciences (CMME, Hospital Sainte-Anne), Institute of Psychiatry and Neuroscience of Paris (INSERM UMR1266), Paris, France. [16]PHLIP Mental Health and Painless Medicine clinic, Toronto, Canada. [17]Sorbonne Université, École normale supérieure, PSL University, CNRS, Laboratoire des Biomolécules, LBM, 75005 Paris, France. [18]LCBPT, Université Paris Descartes, Sorbonne Paris Cité, UMR 8601, CNRS, Paris 75006, France. [19]These authors contributed equally: Mathieu Favier, Elena Martin Garcia, Romain Icick, Camille de Almeida, Joachim Jehl, Rafael Maldonado, Nicolas Pietrancosta, Stéphanie Daumas, Salah El Mestikawy.  ✉e-mail: mathieu.favier@mail.mcgill.ca; salah.elmestikawy@mcgill.ca

