## [Peer Review File · Nature Communications]

The human VGLUT3-pT8I mutation elicits uneven striatal DA signaling, food or drug maladaptive consumption in male mice.Reviewers' comments:

Reviewer #1 (Remarks to the Author):

Nature Communications NCOMMS-23-08286

In this study, Favier et al. report a series of findings related to the physiological and behavioural characterisation of the consequences of the pT8I mutation of the VGLUT3, drawing on epidemiological studies in humans and genetically engineered mice.

This data-heavy manuscript reveals some new and exciting data, such as the asymmetric control of dopamine release in the medial and dorsal territories of the dorsal striatum.

However, I found it challenging to understand the actual take-home message of the manuscript, as I find some results difficult to reconcile. At the same time, the methods are too superficially described/justified, and the results are overall sub-optimally presented.

Major comments:

1. Without being picky, this was the most challenging manuscript to review I have had in about five years, with the outcome of statistical analyses reported in obtuse tables in SOM, and divorced from the figures and results, in the absence of any justification of the statistical tests used (apart from one sentence at the end of the short methods section). As a matter of fact, despite my best intention, I really struggled to understand what analysis was done for what figure. I strongly recommend the outcome of the analyses be included in the figure legends, which should be more explanatory than they are now (very descriptive).
2. How many mice were used overall in this study? How many were deselected? How many were males vs females?
3. Almost a dozen statistical tests have been used in this study, sometimes to interrogate very similar data with no justification. Bonferroni post-hoc tests are used alongside t-tests for multiple comparisons, the former leading to Type II errors while the latter leading to type I errors. The use of non-parametric vs parametric tests is justified in the methods (even though visual inspection of data led me to question the use of t-tests or ANOVAs when data were clearly not normally distributed, i.e., figure S1, 5d, 3i, or suffered from lack of homogeneity of variance/sphericity, i.e., figure S3b). Still, since most conclusions rely on interactions, the authors may want to consider transforming their data so that parametric analyses are carried out throughout.
4. How are the cognitive deficits assessed in the SAPS representative of the severity of cocaine use disorder? (figure 1). Also, with regards to figure 1, how do the patient groups compare to the non-clinical population? Figure 1C it appears that at least three individuals without any variant show higher SAPS scores than the p.T8I carriers. What do the authors make of it?

5. Supporting online table 6 and 7 are the same, and the N column is misleading.
6. The choice of the various behavioural tests and the experimental conditions used should be better justified. For instance, why use both an EPM and maze test? In the EPM, a 10% time spent in the Open Arms reveals conditions that are clearly not conducive to investigations of individual differences. These conditions are usually used to test the anxiolytic properties of specific drugs, as the conditions are clearly highly anxiogenic. But these are not conditions appropriate to study anxiety trait (a 30% time spent in the OA as a baseline is more appropriate in that case as it is not limited by a floor effect).
7. For self-administration, what surgical procedure was carried out to implant the intrajugular catheters in mice? What were the self-administration infusion parameters? How many mice had to be discarded because of catheter failure? What was the discrimination of the animals? The active and inactive nose-pokes must be presented and analysed together in order to establish the qualitative nature of the acquisition of cocaine self-administration. Also, the number of cocaine infusions and intake (in mg/kg) should be presented so the reader can better understand these mutant mice's behaviour.
8. Regarding the reinstatement session (figure 4f), if the reinstatement of seeking behaviour was measured, where are the data showing baseline training of cocaine seeking? (one can only reinstate what has previously been expressed). Here what is being measured is the acquisition of responding for the conditioned reinforcing properties of the cocaine-paired cue in the absence of reinforcement, which eventually results in extinction. To understand the data presented in figure 4j it is necessary to know how long the reinstatement session lasted (reinstatement occurs only for 5 to 10 minutes in mice, then it is extinction that is being measured) and what it is compared to. What was the performance of the animals during the same 5-10 min period of the preceding extinction session? Without these data, it is not possible to interpret this figure.
9. Now, this being said, if the T8I/T8I mice are less likely to acquire cocaine self-administration than wild-type controls (figure 4f), according to an acquisition criterion that remains elusive to the reader, then the take-home message of this figure is that the mutation confers resistance to addiction.... If the individual is less likely to engage in drug use/self-administration, they cannot be deemed to be more vulnerable to developing addiction! Especially when a difference in reinstatement falls short of supporting any addiction-related phenotype on its own.
10. Figure 5. There are several caveats in the results presented in figure 5. First of all, after devaluation, in the absence of a post-extinction consummatory test, it is not possible to state that the persistence of responding in the T8I/T8I group is reflective of habitual control over behaviour: how can the authors be sure the devaluation worked in these animals (since they can ingest more high caloric solution than the wild type animals without showing any satiety effect, as shown figure S3). In addition, figure 5b should be analysed using a 2-way ANOVA, with two conditions (devalued vs non-devalued) and two genotypes as factors, with an interaction opening up subsequent post-hoc comparisons.
11. Figure 5d-k. It seems that the seminal studies that used and validated the 3-criteria model of addiction are not cited in this manuscript.
12. Surprisingly, most of the mice in this study did not develop high levels of responding during the no-reward periods or showed a high level of motivation. Could it be the case that the experimental conditions are not adequate properly to identify 3-criteria individuals (which is clearly the case in the WT

population figure 5g)? What is the rationale for using a 75th percentile threshold for the determination of a positive score for one criterion? The number of foot shocks received (figure 5f) does not reflect compulsivity: how many rewards did the animals receive under punishment compared to baseline?

13. Finally, Figures 5g and 5h are misleading as the identification of 0 vs 3-crit individuals is based on the distribution of each population. In order to establish if the T8I/T8I variant confers increased vulnerability to develop addiction-like behaviour, both WT and mutant animals should be used in the same analysis, and the genotype of the individuals belonging to the 0crit and 3crit populations should be retrospectively investigated. It remains to be justified why an individual that does not meet three criteria threshold for addiction (e.g. a 2-crit individual) is deemed addicted in this study.

14. Figure S3: it appears that only 2 individuals contribute to the tenuous difference that is reported on panel d (one individual in the T8I/T8I group being above the rest of the population and one individual in the WT group being below the rest of the population. Can the authors better discuss this? Also, it may be worth comparing directly how much sucrose vs water an individual drank during the binge test instead of running the two comparisons independently (e.g., compare S3b to S3d). Figure S3i, is there really a time x group interaction there?

15. The comparisons of the incidence of the genotype in human studies may be better supported by Bayesian statistics than the exact test used here, considering that there is overall no asymmetric distribution of genotypes in the clinical and non-clinical populations.

Wrap-up statement:

Here the results suggest that the same SNIP contributes to an exacerbation of the vulnerability to develop an addiction to cocaine, anorexia and binge eating of highly palatable food. So these compulsive disorders should be more common together in humans than with other co-morbid elements. What is the evidence it is the case? In addition, SNIP carriers are shown here to be less likely to self-administer cocaine than wild-type individuals, the 3-criteria model of addiction was not used for cocaine and was not adequately tailored to food. In addition, the results, as presented, do not support the conclusion that the SNIP confers greater vulnerability to develop compulsive food intake (figure 5f), while the effect of SNIP on actual body weight loss in the ABA model is tenuous (figure S3).

Finally, the suggestion that the higher vulnerability to develop compulsivity is due to a facilitation of habit formation is at odds with the literature suggesting it is not the speed of habit formation but the ability to disengage the habit system in the face of adverse consequences that confers the vulnerability to develop compulsivity.

Reviewer #2 (Remarks to the Author):

A missense polymorphism, VGLUT3-p.T8I, was identified in patients with substance use disorders (SUDs) and eating disorders (EDs). A mouse line was generated to understand the neurochemical and behavioral impact of the p.T8I variant. In VGLUT3T8I/T8I

mice, glutamate signaling was unchanged but vesicular synergy and ACh release were blunted. Mutant mice exhibited an uneven dopaminergic transmission in the dorsal striatum facilitating habit formation and exacerbating maladaptive use of drug or food. Increasing Ach tone with donepezil reversed the self-starvation phenotype observed in VGLUT3T8I/T8I mice. The authors conclude that unbalanced dopaminergic transmission in the caudate vs putamen could be a common mechanism between SUDs and EDs and advocates for the use of donepezil to treat these pathologies. This is potentially an important study and exciting finding with elegant and impressive basic science. However, the claim that this polymorphism offers insight into anorexia nervosa seems to be questionable, and needs additional data.

A careful reading of the Bahji paper confirms what is well known in the ED literature and clinical observation. That is (see Page 59) “binge eating/purging behaviors are associated with higher frequency of substance abuse. It should be noted that there tends to be 2 subtypes of AN. Those that restrict and those that binge/purge. And the literature has been consistent (See page 51) “SUD significantly higher in binge-purge group than in the restrictive group. (And see Holderness IJED 1994)

The paper does not appear to describe the rate of this polymorphism in those with AN or subtypes of AN. They do note that in the extended clinical sample (ED + SUD samples) this polymorphism was found in 8 of 632 cases and this was not different than the controls. They include a case of a person with bulimia nervosa and cocaine abuse. Could the data be analyzed to compare restrictor-type AN to binge-purge ED? It may be that the polymorphism contributes to some shared behavior such as emotional or impulse dysregulation common to both SUD and bulimia. Alternatively, is there data to support the possibility that this polymorphism contribute to some behavior shared by restrictor AN and BN-AN? It should be noted that some AN also restrict and purge. Does this subgroup express this polymorphism and if so, are they more similar to restricting or binge-purge AN?

The role of habit in anorexia nervosa remains uncertain as those with AN do not show a generalized deficit in the balance between goal-directed and habitual control of behavior. (see Godier Psych res 244:214-22, 2016). Alternatively, the role of this polymorphism in anxiety should be considered, as measures of anxiety are elevated in AN independent of state of the illness, and measures of anxiety have been shown to be associated with dorsal striatum dopamine function in AN (Frank Biol Psychiatry PMID 15992780 2005) and anxiety is genetically transmitted in AN and occurs in childhood before the onset of AN symptoms (Meier Eur Eat Disord Rev 2015, PMID: 26347124).

In summary, these findings could be very important in providing insight into physiological mechanisms in ED but it would be useful to know whether restrictor-type and bulimic-type AN both have this alteration

or whether it is specific for bulimic type behaviors (including those with bulimia nervosa) but not anorexia nervosa, given the relationship of bulimia and SUD. The title should be updated to reflect new findings as it seems premature to describe this as a model for AN.

Reviewer #3 (Remarks to the Author):

The manuscript by Favier et al reports on a comprehensive study pinpointing the subtle but important roles played by VGLUT3 expressed by cholinergic interneurons in striatum. Their approach was to study the naturally occurring population of humans with intronic SNP in VGLUT3 at p.T8I, a SNP mutation of the gene in mice (VGLUT3-pT8I) and rescue of (and failure to rescue) VGLuT3-null mice with the VGLuT3-pT8I gene. Their methods spanned from electrophysiology in cultured neurons to a wide battery of behavioral tests, in vitro neurotransmitter release/uptake tests done to synaptic vesicles collected from VGLuT3 null mice, in vivo analyses of VGLUT3 null mice rescued with VGLuT3-pT8I (threonine in position 8, not present in VGLuT1 or 2, mutated to isoleucine), fiber photometry and in vivo voltammetry in striatum to measure ACh release and dopamine (DA) release of WT and VGLuT3T8I/T8I mice, 3D structural analysis of VGLuT3, and a variety of immunocytochemical analyses to assess potential alteration in expression caused by T8I mutation. They report on many data that indicate unperturbed molecular, synaptic and behavioral phenotypes and a handful of findings that could only be appreciated by deeper analyses. The phenotypes that they discovered to be perturbed by VGLuT3 mutations are as follows:

- 1) vesicular synergy (glutamate-dependent ACh uptake Fig 3i; ACh release in Fig 3m-n) are blunted in VGLuT3T8I/T8I mice (N= 7).
- 2) DA transmission is uneven (reduced DA release in DMS but not in DLS) in VGLuT3T8I/T8I mice, (N= 14, Fig 3)
- 3) Enhanced cue-induced vulnerability to cocaine (N=5, Fig 4f-j)
- 4) enhanced habit formation in sucrose seeking (N=12, Fig. 5b) and binge-like drinking of high-sucrose solution during the 1st hour (Fig. 5m, N=10)
- 5) self-starvation that was more severe for the VGLuT3T8I/T8I mice(N=10) and reversal with donepezil (N=8)

6) p.T8I variant of VGluT3 (1.3%, heterozygous, N=8, N=3 for cocaine-addicted patients in Fig 1c) that is associated with higher incidence of psychotic symptoms and SUD (substance use disorder).

These are each very interesting, important, impactful and well-executed as a combination of findings, providing insights into the molecular mechanisms underlying maladaptive behaviors associated with SUD and anorexia nervosa, with implications for pharmacological treatments of both illnesses. The only shortcoming to the entire study is the small sample size of the humans with SNP in threonine 8 (N=3) and also of the animals pointing to the enhanced cue-induced vulnerability to cocaine (N=5). They also comment on ONE PATIENT with co-morbidity of eating disorders and drug abuse to suggest that “p.T8I may be associated with both SUDs and EDs”. However, since statistical analyses indicate significances and all parts combine to tell a cohesive story pointing to enhanced habit formation involving VGluT3 expressed by cholinergic interneurons in dorsomedial striatum, I believe all data should be accepted as presentable.

Minor points:

With regard to finding #2 in the list above, the word ‘Uneven,’ which appears in the Abstract was unclear. What I spelled out in parenthesis is clearer (is this what authors meant?), although more wordy.

Page 8. Authors report that VGluT3 activity accounts for 43% of total striatal vesicular uptake of glutamate. Does this not seem very high, considering the density of cholinergic axons versus glutamatergic axons in striatum? Please comment and relate the new findings to precedents, if there are any.

Page 9. Why was KCl-depolarization used to evoke DA release? Wouldn't nicotine or glutamate have been better ways to probe for “ACh-glutamate co-transmission ... in controlling DA release”? Please provide a brief rationale for the method used to evoke DA release.

The cited reference #66 describing the method used to assess habit formation in obtaining sucrose pellets under devalued versus valued condition was difficult to find. Please add information about the volume and issue to this reference.

Point-by-point answers to Reviewer comments

We would like to express our gratitude to Reviewers for their in-depth reading of the manuscript as well as for their comments and constructive suggestions. To prepare this new version of the manuscript, we considered all their comments.

Thanks to these changes, the new version of the manuscript is both clearer and more convincing.

Our responses to queries of the Reviewers as well as modifications made to the text are in blue.

Answers to Reviewer #1:

In this study, Favier et al. report a series of findings related to the physiological and behavioural characterisation of the consequences of the pT8I mutation of the VGLUT3, drawing on epidemiological studies in humans and genetically engineered mice. This data-heavy manuscript reveals some new and exciting data, such as the asymmetric control of dopamine release in the medial and dorsal territories of the dorsal striatum. However, I found it challenging to understand the actual take-home message of the manuscript, as I find some results difficult to reconcile. At the same time, the methods are too superficially described/justified, and the results are overall sub-optimally presented.

We apologize for the complexity and lack of clarity of the initial manuscript. In this new version we tried to take into account all recommendations from Reviewer #1 to improve the manuscript.

Major comments:

1. Without being picky, this was the most challenging manuscript to review I have had in about five years, with the outcome of statistical analyses reported in obtuse tables in SOM, and divorced from the figures and results, in the absence of any justification of the statistical tests used (apart from one sentence at the end of the short methods section). As a matter of fact, despite my best intention, I really struggled to understand what analysis was done for what figure. I strongly recommend the outcome of the analyses be included in the figure legends, which should be more explanatory than they are now (very descriptive).

We thank Reviewer #1 for raising this important point. As recommended, we tried to make this version of the manuscript more reader friendly. Names of statistic tests and p values are in Figure legends and in Results. We also provide more explanation to legends.

2. How many mice were used overall in this study? How many were deselected? How many were males vs females?

In this study 121 WT mice and 114 VGLUT3^{T8I/T8I} mice were used. The total number of mice used is indicated in supplemental statistic tables and in Material and Methods (page 20 lines 618-619). We used only male since we documented in a previous study that striatal hypocholinergic phenotypes are similar in male and female (Favier et al. JCI 2020 PMID: 33164988). This is now indicated in material and methods (page 20, lines 619-621). Furthermore, we currently replicated part of the study with females and found no differences with males (not shown).

No animals were excluded, except:

- In the cocaine-self administration test where mice that did not reached the criteria of operant conditioning learning (Fig. 4g, 58 % for WT mice and 29% for VGLUT3^{T8I/T8I}) following FR1 and FR3 training were excluded from following experiments (PR, extinction and cue-induced relapse),
- In the sensory devaluation paradigm where mice that did not consume a minimum of 0.4 g.

3. Almost a dozen statistical tests have been used in this study, sometimes to interrogate very similar data with no justification. Bonferroni post-hoc tests are used alongside t-tests for multiple comparisons, the former leading to Type II errors while the latter leading to type I errors. The use of non-parametric vs parametric tests is justified in the methods (even though visual inspection of data led me to question the use of t-tests or ANOVAs when data were clearly not normally distributed, i.e., figure S1, 5d, 3i, or suffered from lack of homogeneity of variance/sphericity, i.e., figure S3b). Still, since most conclusions rely on interactions, the authors may want to consider transforming their data so that parametric analyses are carried out throughout.

We agree with Reviewer #1 that many different statistical tests were used in this study. This is primarily due to the fact that a lot of different type of data were analyzed, with several paradigms used depending on experiments.

As suggested by Reviewer #1 (see point 15 below), we used Bayesian Wilcoxon tests to analyze Fig. S1 (main text, page 7, lines 168-174 and Supplementary Material, page 16, lines 185-195 and Supplementary Fig. 1 page 19).

For Fig. S2 (fiber photometry), data were analyzed with a non-parametric test Rank sum test. For vesicular uptake data (Fig 3e), we used a Kolmogorov-Smirnov to validate normality followed by one-way ANOVA and Tukey's post hoc test (main text, page 9, lines 233-235 and 245-246).

For persistence of the response in food addiction we used for Fig. 5c mixed effect model (restricted maximum likelihood, REML) (main text page 13, lines 391-392) and for Fig. 5d, a non-parametric Mann-Whitney test ($U=67$, $p=0.2088$) (main text page 13, lines 398-399).

4. How are the cognitive deficits assessed in the SAPS representative of the severity of cocaine use disorder? (figure 1).

The severity of substance use disorders is a complex, multi-dimensional concept. Here, we used the SAPS score as a proxy for such severity. The SAPS measures psychotic and motor symptoms that are thought to represent both a consequence of heavy cocaine use (in interaction with individual vulnerability) and a cause of specific burden, due to paranoid delusions or agitation (see, e.g., PMID: 32866679 and 33159041). Thus, for clinicians, the SAPS represents a partial, yet useful assessment of the severity of CUD. For example, SAPS total score correlates highly with the number of DSM-IV criteria for CUD in our sample ($\rho = 0.41$, $p = 10^{-6}$) as shown in the additional Figure below (not included in the submitted manuscript). Several sentences relative to this point have been added in the Supplementary Methods (Suppl material, page 16, lines 171-184).

Figure for Reviewers. Correlation SAPS/DSM-IV criteria in our sample of CUD patients

Importantly, in our study, the association of SAPS with carriers of the p.T8I variant remained significant after covarying for other factors influencing SAPS-total score. These comments are added to Results (page 7, lines 160-168).

Also, with regards to figure 1, how do the patient groups compare to the non-clinical population?

The comparison to non-clinical controls has not been performed in this study. Here, we compared patients with the p.T8I variant, to other VGLUT3 variants or no variant within an outpatient sample seeking treatment for substance use disorders (alcohol, cannabis,

cocaine, opioid). Moreover, the SAPS cannot be determined in a control population of non-cocaine users.

Fig 1C it appears that at least three individuals without any variant show higher SAPS scores than the p.T8I carriers. What do the authors make of it?

All carriers of p.T8I carriers have high scores on the SAPS scale. However, all patients with high SAPS scores are not p.T8I variants carriers.

In Fig. 1c, the left bar represents a cohort of patients with severe CUD but no VGLUT3 variant. It is therefore normal that some individuals have high SAPS scores.

Numerous biological and environmental factors are likely to play a role in psychotic symptoms of CUD patients. This could reflect the multifactorial nature of experiencing psychotic symptoms after cocaine use (see PMID: 32866679). The three outliers in the non-T8I group reinforce the confidence in the significantly higher median score in T8I carriers. Indeed, removing the outliers in the non-T8I group would reinforce the association with T8I. As listed in Supplementary Data, median SAPS scores were 6 for patients without any mutation vs. 11 for T8I carriers.

5. Supporting online table 6 and 7 are the same, and the N column is misleading.

Mistakes in Supplementary Table S6 (page 20) and Table S7 (page 21) have been corrected.

6. The choice of the various behavioural tests and the experimental conditions used should be better justified. For instance, why use both an EPM and maze test? In the EPM, a 10% time spent in the Open Arms reveals conditions that are clearly not conducive to investigations of individual differences. These conditions are usually used to test the anxiolytic properties of specific drugs, as the conditions are clearly highly anxiogenic. But these are not conditions appropriate to study anxiety trait (a 30% time spent in the OA as a baseline is more appropriate in that case as it is not limited by a floor effect).

We previously established that VGLUT3 loss of function leads to anxiety (Amilhon et al. J Neurosci 2010, PMID: 20147547). Since one aim of this study was to establish whether the p.T8I mutation results in a VGLUT3-loss or gain of function we explored anxiety behaviors of VGLUT3^{T8I/T8I} mice in several different paradigm routinely used in the lab: EPM (Fig. 4b-d), marble-burying (Fig. 4e), O-maze (Fig. S3). In this revised version, we also added the dark/light chamber test (Fig S3c,e).

As pointed by Reviewer #1, conditions used in the EPM are relatively stringent. However, they allow a direct comparison with our previous investigation with other VGLUT3 mutants (VGLUT3-KO in Amhilon et al. J Neurosci 2010, PMID: 20147547 and VGLUT3^{A211V/A211V} in Ramet et al., J Neurosci 2017 PMID: 28314816). All four anxiety-related tests (EPM, O-maze, marble burying and dark/light box) point to an absence of excessive anxiety behaviors of VGLUT3^{T8I/T8I} mice. This is an important finding since anxiety could be a confounding factor for several test used in our investigations.

7. For self-administration, what surgical procedure was carried out to implant the intrajugular catheters in mice? What were the self-administration infusion parameters? How many mice had to be discarded because of catheter failure? What was the discrimination of the animals? The active and inactive nose-pokes must be presented and analysed together in order to establish the qualitative nature of the acquisition of cocaine self-administration. Also, the number of cocaine infusions and intake (in mg/kg) should be presented so the reader can better understand these mutant mice's behaviour.

To clarify the cocaine self-administration experiments, we ran an additional cohort of mice, and we now present more robust data with n=14 WT mice and n=11 VGLUT3^{T8I/T8I} mice (see comments below).

- The surgical procedure and all experimental conditions are now included in Supplementary Methods (pages 16-18, lines 197-265).
- We added a graph showing the number of inactive nosepokes (Fig. S3h). These data illustrate that all animals discriminate between active and inactive lever. These findings are commented in Results (page 11, lines 321-322). Additionally, we also included in Results a sentence regarding the discrimination between active and inactive levers (page 11, lines 324-325).
- We included in the supplementary material an additional graph showing the number of infusions (left Y-axis) and the intake in mg/kg/infusion of 23 μ l (right Y-axis) (Fig. S3g). These data are commented in Results (page 11, lines 329-336).

8. Regarding the reinstatement session (figure 4f), if the reinstatement of seeking behaviour was measured, where are the data showing baseline training of cocaine seeking? (one can only reinstate what has previously been expressed). Here what is being measured is the acquisition of responding for the conditioned reinforcing properties of the cocaine-paired cue in the absence of reinforcement, which eventually results in extinction. To understand the data presented in figure 4j it is necessary to know how long the reinstatement session lasted (reinstatement occurs only for 5 to 10 minutes in mice, then it is extinction that is being

measured) and what it is compared to. What was the performance of the animals during the same 5-10 min period of the preceding extinction session? Without these data, it is not possible to interpret this figure.

Reinstatement data were included in the first version of this manuscript in Fig. S3. To further clarify the behavioral extinction, we now included the baseline values of the acquisition and the extinguished behavior just before the reinstatement session in Fig. 4k. These results are commented (pages 11-12, lines 334-340).

Extinction sessions lasted 2h similar to the duration of the operant training in FR1 and FR3 and equal to the cue-induced reinstatement session. In the supplementary material, we have included a figure showing last day of extinction and the first day of reinstatement separated by blocks of 10 minutes (Fig S3h and i). These panels are commented in Results (page 12, lines 352-362, Fig S3h,i).

9. Now, this being said, if the T8I/T8I mice are less likely to acquire cocaine self-administration than wild-type controls (figure 4f), according to an acquisition criterion that remains elusive to the reader, then the take-home message of this figure is that the mutation confers resistance to addiction.... If the individual is less likely to engage in drug use/self-administration, they cannot be deemed to be more vulnerable to developing addiction! Especially when a difference in reinstatement falls short of supporting any addiction-related phenotype on its own.

Acquisition criteria are now clearly explained in the methods section of the supplementary material in the paragraph titled: "*Experimental design of the model of reinstatement of operant conditioning maintained by cocaine*" (Supplementary Material, pages 18, lines 240-265).

It is important to underline that the acquisition of an operant self-administration behavior is directly related to the reinforcing properties of a drug, which is associated with drug-seeking behavior (Werner et al. 2021 PMID: 33066961). Drug-seeking behavior is just the first step to develop, or not, later addictive-like behaviors (Shaham et al. 2003 PMID: 12402102). To evaluate behavioral parameters specific to the development of drug addiction it is mandatory to assess additional operant behavioral responses more directly related to the development of this behavioral disorder, such as reinstatement to drug-seeking after extinction (Belin et al., 2008 PMID: 18535246). In our experimental conditions, mutant mice and WT mice differ in the acquisition of operant conditioning maintained by cocaine. However, during the FR3 period and especially at the end of the training period, both genotypes reached the same

level of cocaine intake (Fig 4f). Following FR3 acquisition extinction levels were similar between the 2 genotypes.

Furthermore, as noticed by Reviewer #1, global consideration FR1 and FR3 suggest that WT mice have a general trend to self-administer more cocaine than VGLUT3^{T8I/T8I} mice (at least at the beginning of the acquisition). These data support the idea that the increased vulnerability of VGLUT3^{T8I/T8I} mice for cocaine-induced reinstatement is specifically related to addiction and not to differences in sensitivity to reinforcing properties of the drug between genotypes.

For these reasons, we can assume that the acquisition levels do not influence the levels of reinstatement.

10. Figure 5. There are several caveats in the results presented in figure 5. First of all, after devaluation, in the absence of a post-extinction consummatory test, it is not possible to state that the persistence of responding in the T8I/T8I group is reflective of habitual control over behaviour: how can the authors be sure the devaluation worked in these animals (since they can ingest more high caloric solution than the wild type animals without showing any satiety effect, as shown figure S3). In addition, figure 5b should be analysed using a 2-way ANOVA, with two conditions (devalued vs non-devalued) and two genotypes as factors, with an interaction opening up subsequent post-hoc comparisons.

The sensory devaluation test used in the present study to dissociate goal-directed behavior from habitual behavior is commonly used and has been validated in our previous publication (Favier et al., 2020, J Clinical Invest, PMID: 33164988) as well as by several other teams (for instance see: Rossi MA, Yin HH. Methods for studying habitual behavior in mice. Curr Protoc Neurosci Chapter 8, Unit 8 29 (2012). Bradfield et al., Neuron 2013 PMID: 23770257).

As described in Material and Methods, all mice were given for 1h *ad libitum* access to home chow (valued condition) or sucrose pellets (devalued condition). The amount of food consumed during the *ad libitum* session was recorded, and mice that did not consume a minimum of 0.4g of each food were not included in the analyses. This process allows all mice, under both valued and devalued conditions, to be satiated prior to the extinction test, ensuring that differences in performance in the subsequent extinction test is specifically related to the decrease in reward value.

Relatively to the comment on the satiety effect and Fig. S4 (former Fig. S3), it is important to dissociate between general satiety effect and sensory-specific devaluation. Indeed, in the binge model, the 1-hour chow preload test after 16 consecutive days of procedure decreased the subsequent consumption of regular home chow for both WT mice and VGLUT3^{T8I/T8I} mice (see Figure below). Consequently, the fact that VGLUT3^{T8I/T8I} mice still binge more sucrose

Figure for Reviewers. Chow consumption

solution than WT mice after chow preload is most probably related to exacerbated habitual behavior of mutant mice. This finding is well in line with our proposal that increased habits in VGLUT3^{T8i/T8i} mice confers vulnerability to develop binge-like sucrose overconsumption (even if the binge model does not intrinsically allow to formally dissociate goal-directed behaviors from habits).

As suggested by Reviewer #1, Fig. 5b is now analyzed using a 2-way ANOVA and added to the statistical tables in Supplementary Material, revealing both a general value effect and a significant interaction between genotype and conditions (valued vs devalued) (see page 13, lines 382-383).

11. Figure 5d-k. It seems that the seminal studies that used and validated the 3-criteria model of addiction are not cited in this manuscript.

This mistake has been corrected (page 12, line 370-371).

12. Surprisingly, most of the mice in this study did not develop high levels of responding during the no-reward periods or showed a high level of motivation. Could it be the case that the experimental conditions are not adequate properly to identify 3-criteria individuals (which is clearly the case in the WT population figure 5g)? What is the rationale for using a 75th percentile threshold for the determination of a positive score for one criterion? The number of foot shocks received (figure 5f) does not reflect compulsivity: how many rewards did the animals receive under punishment compared to baseline?

We agree with Reviewer #1 that the number of responses in this experiment was very low. We believe that this could be related to the C57BL/6N genetic background of mice corresponding to a phenotype of low responders associated with increased anxiety-like behavior. Conditions used herein were adequate since they were similar for the whole population. The rationale for using the percentile 75th as a threshold is to have a qualitative measure with sufficient mice separating the top 25% cluster as a vulnerable phenotype positive for one criterion. The number of shocks received reflects compulsivity since mice responds to the lever, although it is associated with negative consequences and cannot stop.

As can be expected, the number of rewards obtained during a shock session is significantly lower than baseline, due to the punishment associated with the reinforcer.

13. Finally, Figures 5g and 5h are misleading as the identification of 0 vs 3-crit individuals is based on the distribution of each population. In order to establish if the T8I/T8I variant confers increased vulnerability to develop addiction-like behaviour, both WT and mutant animals should be used in the same analysis, and the genotype of the individuals belonging to the 0crit and 3crit populations should be retrospectively investigated.

The procedure to evaluate food addiction in mice was previously validated by several publications from the Maldonado's laboratory (see for example, Domingo-Rodriguez Nature Comm 2020 PMID 32034128, Martín-García Bio Protoc 2020 PMID 33659433, García-Blanco J Clin Invest 2022 PMID 35349487).

We agree with Reviewer #1 that identifying 0 vs. 3 crit individuals based on the distribution of each population could be confusing. This is not what is done in the present study. The attribution of addiction criteria is based on the WT reference group. To clarify this important point, a sentence in "Methods" section states that the threshold to attribute positive criteria is considered from the percentile 75 of the WT group (page 27, lines 888-890).

Regarding the retrospective analysis of addicted and not addicted mice from each genotype: no statistical differences were obtained among groups (see interaction between genotype and addiction criteria, Supplementary Table 5, food addiction page 11). We therefore represent only data regarding genotypes. A sentence from the manuscript specifies that addicted mice from each genotype behaves similarly and that no interaction between genotype and addiction was found (pages 13, lines 396-401).

Neurobiological mechanisms underlying the three addiction criteria differ anatomically: the medial prefrontal cortex (mPFC) is pivotal for compulsivity and persistence whereas the mesolimbic pathway is central for motivation (Maldonado et al. 2021 PMID: 33482225). Importantly, in line with the diagnosis of addiction in the DSM-5 for substance use disorders, or the YFAS 2.0 for food addiction, mice that are above the threshold of 2 criteria are also considered as addicted. As shown by our previous publications, it is usual to have a group of addicted mice with only two criteria of addiction.

It remains to be justified why an individual that does not meet three criteria threshold for addiction (e.g. a 2-crit individual) is deemed addicted in this study.

As previously reported by the Maldonado team, the use of both 2 and 3 positive criteria to consider a mouse as addicted to food was previously justified (Mancino et al., 2015

Neuropsychopharm PMID: 25944409) in order to obtain a percentage of animals reaching these criteria similar to humans that reach the DSM-4 and DSM-5 criteria of drug addiction and substance use disorder, respectively. The same criteria (2 and 3) were also considered in more recent studies that evaluated food addiction using the same operant paradigm (Domingo-Rodriguez Nature Comm 2020 PMID 32034128, Garcia-Blanco J Clin Invest 2022 PMID: 35349487). In both studies, the percentage of mice reaching food addiction was similar to those above-mentioned in humans.

14. Figure S3: it appears that only 2 individuals contribute to the tenuous difference that is reported on panel d (one individual in the T8I/T8I group being above the rest of the population and one individual in the WT group being below the rest of the population. Can the authors better discuss this?

We are not sure to fully understand the point raised by Reviewer #1 about Fig. S4d (former Fig. S3d). The sucrose binge test is designed to model bulimia: ie consumption of large amount of food in a relatively brief time. This is why analysis of the data from the 4h test distinguishes the first hours (H0-H1) from the following 3 hours (H1-H4). As can be seen in Fig S4d (former Fig. S3d) if a line (spotted red line, see Figure for Reviewers) is drawn at the

level of the highest point of WT mice, it is clear that a majority of mutant mice (9 out of 10) performed at or above the dotted red line. The significant difference observed between WT mice and mutant mice during the 4hr is driven solely by the first hour since no difference are observed during the final 3hr (H1-H4).

Figure for Reviewers extracted from Fig S4d.

This point is discussed in Results (page 14, lines 432-436).

Also, it may be worth comparing directly how much sucrose vs water an individual drank during the binge test instead of running the two comparisons independently (e.g., compare S3b to S3d). Figure S3i, is there really a time x group interaction there?

Again, we are not sure to fully understand the point raised by Reviewer #1 about panels S3b-d. Water intake was not measured during the 1st hour (H0-H1) and therefore cannot be

compared with sucrose intake for this period. All individuals from both genotypes consumed much more sucrose than water, as expected in this binge-like sucrose overconsumption test. Concerning comments related to Fig. S4i (former Fig. S3i), we confirmed that there is no time x genotype interaction (Supplementary Table S13).

15. The comparisons of the incidence of the genotype in human studies may be better supported by Bayesian statistics than the exact test used here, considering that there is overall no asymmetric distribution of genotypes in the clinical and non-clinical populations.

We thank Reviewer #1 for suggesting this additional statistical approach to our study. Bayesian Wilcoxon tests were performed using JASP software (main text, page 7, lines 168-174 and Supplementary Material, page 16, lines 185-195 and Fig. S1b page 19). These additional analyses show that Bayes factor support an actual difference in cocaine-induced psychosis (SAPS scores), corresponding to a four-fold higher likelihood that these scores were higher in p.T8I carriers than in individuals with no mutation of *SLC17A8* compared to the opposite hypothesis (Legend of Fig. S1b, page 19, lines 277-284). We hope that these non-frequentist analyses, which are often overlooked, will appeal to readers, and contains all the required information for interpretation.

Wrap-up statement:

Here the results suggest that the same SNIP contributes to an exacerbation of the vulnerability to develop an addiction to cocaine, anorexia and binge eating of highly palatable food. So these compulsive disorders should be more common together in humans than with other co-morbid elements. What is the evidence it is the case?

The point raised by the Referee is well taken and important. Several reviews show a frequent association between EDs and SUDs (Devoe et al., 2021 PMID: 34895358, Conason et al. 2006 PMID: 16497847). Metanalysis even report a 23-37% comorbidity between EDs and SUDs. SUDs seem to associate more often with bulimia or with binge-purge anorexia (AN-BP) than with restrictive type anorexia (AN-R). Interestingly we found one bulimic and cocaine addict patient carrying the T8I variant.

Concerning compulsion, more than 50% of anorexic patient also have obsessive-compulsive disorder (Pollack et al., 2013 PMID 23557823, Altman et al., 2009 PMID: 19744759). Interestingly, we recently obtained preliminary evidence that *VGLUT3*^{T8I/T8I} mice are more vulnerable to compulsive grooming in the splash test. Furthermore, this compulsive grooming of mutant mice is abolished by donepezil. This finding suggests that, in rodent, a lower striatal cholinergic tone could drive pathological grooming.

Figure for Reviewers. Compulsive grooming of VGLUT3^{T81/T81} mice and reversal by donepezil. Note that it takes 5 days before obtaining a significant difference (El Mestikawy unpublished).

These unpublished splash test data combined with data from the present report suggest that the p.T81 variant confers a vulnerability to mice model of compulsive grooming, addiction-like, anorexia-like and bulimia-like behaviors. These observations led us to propose that a decreased dopamine signaling in the caudate/dorsomedial striatum relatively to the putamen/dorsolateral striatum could be a core factor triggering several compulsive behaviors (as shown in the model Supplementary Figure 6).

In addition, SNIP carriers are shown here to be less likely to self-administer cocaine than wild-type individuals, the 3-criteria model of addiction was not used for cocaine and was not adequately tailored to food. In addition, the results, as presented, do not support the conclusion that the SNIP confers greater vulnerability to develop compulsive food intake (figure 5f), while the effect of SNIP on actual body weight loss in the ABA model is tenuous (figure S3).

The point raised by Reviewer #1 is correct, mutant mice undergoing the ABA model demonstrate a limited weight loss relatively WT littermate. However, when the threshold “animal with less than 75% bw” is used a highly significant difference is observed between WT and mutant mice (Fig. 5o, Log-rank (Mantel-Cox) post hoc comparison $p < 0.001$, Gehan-Breslow-Wilcoxon post hoc comparison $p < 0.001$, Kaplan-Meier test). This criterion evaluates how many animals on a given day will reach this critical threshold. This is a central parameter not only in the well-validated ABA rodent model but also in the clinic where patients below this threshold are considered to be in severe pathological status. We previously used the well-validated ABA model, to establish that another hypocholinergic mouse line (VACHTcKO mice) is more vulnerable to self-starvation than WT mice. Interestingly, we observed “tenuous” but significant differences between WT mice and VACHTcKO mice in terms of body weight whereas marked differences were obtained on

decreased food consumption (compared to baseline for each mouse) and critical weight loss (more than 25% of baseline BW) between the two genotypes.

Finally, the suggestion that the higher vulnerability to develop compulsivity is due to a facilitation of habit formation is at odds with the literature suggesting it is not the speed of habit formation but the ability to disengage the habit system in the face of adverse consequences that confers the vulnerability to develop compulsivity.

In agreement with the comment from Reviewer #1, the classic sensory devaluation test used herein assess the ability of mice to disengage from a previously learned behavior (FR1 obtention of sucrose pellets). Indeed, even when VGLUT3^{T8I/T8I} mice had previously free access to ad libitum sucrose pellets (devalued condition) they persist in nose poking for sucrose pellets. Interestingly, this persistent behavior of VGLUT3^{T8I/T8I} mice is reminiscent of the lack of behavioral flexibility observed with VAcHTcKO, another mouse line with reduced striatal cholinergic signaling (see Favier et al., JCI 2020 PMID 33164988). We agree with Reviewer #1 that the ability to disengage the habit system is the key parameter to protect mice (and humans) from compulsive self-destructive behaviors. Based on this recommendation, we changed a sentence in the manuscript (page 12, lines 372-373).

Reviewer #2 (Remarks to the Author):

A missense polymorphism, VGLUT3-p.T8I, was identified in patients with substance use disorders (SUDs) and eating disorders (EDs). A mouse line was generated to understand the neurochemical and behavioral impact of the p.T8I variant. In VGLUT3T8I/T8I mice, glutamate signaling was unchanged but vesicular synergy and ACh release were blunted. Mutant mice exhibited an uneven dopaminergic transmission in the dorsal striatum facilitating habit formation and exacerbating maladaptive use of drug or food. Increasing Ach tone with donepezil reversed the self-starvation phenotype observed in VGLUT3T8I/T8I mice. The authors conclude that unbalanced dopaminergic transmission in the caudate vs putamen could be a common mechanism between SUDs and EDS and advocates for the use of donepezil to treat these pathologies. This is potentially an important study and exciting finding with elegant and impressive basic science. However, the claim that this polymorphism offers insight into anorexia nervosa seems to be questionable, and needs additional data.

This critical point raised by Reviewer #2 is well taken. We found “only” 8 addicts (Paris SUDs samples) and 1 bulimic/addict patient (Montreal EDs sample) but no AN patient. Ethnicity is a well-established risk factor for eating disorders (see Cheng et al. PMID: 30529736). As mentioned in the manuscript, the p.T8I variant is observed essentially in patients with African origin. According to the review from H.W. Hoek (2016, PMID 27608181) the prevalence of AN in Africans is <0.01% compared to 1% in Caucasians and Asians. Patients carrying the p.T8I variant account for 2% of the general population and less than 5-10% of mental health African patients (based on the evidence presented in this manuscript). The fact that individuals with African origin are much less likely to be diagnosed with AN could explain why we did not find anorexic p.T8I carriers. Identifying p.T8I carriers with AN or either restrictive (AN-R) or binge-purge (AN-BP) subtype would necessitate scanning very large DNA banks of African ED patients.

in Africans, the prevalence of bulimia nervosa (BN) and binge eating disorders (BED) (0.87% and 4.45% respectively) are more frequent than AN (0.01%). Interestingly we found one BN patient in the ED sample. Overall, these observations suggest that AN-BP harboring the p.T8I variant could be more frequent than AN-R. However, this remains to be documented in large sample of African anorexic patients.

Nonetheless, it is important to mention that VGLUT3^{T8I/T8I} mice are equally vulnerable to sucrose binge eating (bulimia-like behavior) as to self-starvation in the ABA model (anorexia-like behavior). This observation points to important “social” differences between rodent and humans as well as to the limits of animal models.

These interesting points raised by Reviewer #2 are now discussed in the Discussion (page 16-17 lines 492-509).

VGLUT3^{T8I/T8I} mice clearly demonstrate a vulnerability to the ABA model and furthermore self-starvation phenotypes are reversed by donepezil. Interestingly in collaboration with psychiatrists 9 ED patients including (AN-R and AN-BP) were successfully treated with low doses of donepezil (case series manuscript in preparation). Therefore, despite the fact that we were not able to identify AN patients carrying the p.T8I mutation, our present data combined with our preliminary unpublished clinical observation support the model proposed in Fig. S6.

A careful reading of the Bahji paper confirms what is well known in the ED literature and clinical observation. That is (see Page 59) “binge eating/purging behaviors are associated with higher frequency of substance abuse. It should be noted that there tends to be 2 subtypes of AN. Those that restrict and those that binge/purge. And the literature has been consistent (See page 51) “SUD significantly higher in binge-purge group than in the restrictive group. (And see Holderness IJED 1994)

This point and the proposed references are now included in the Discussion (page 16 lines 472-476 and lines 500-504)

The paper does not appear to describe the rate of this polymorphism in those with AN or subtypes of AN. They do note that in the extended clinical sample (ED + SUD samples) this polymorphism was found in 8 of 632 cases and this was not different than the controls. They include a case of a person with bulimia nervosa and cocaine abuse. Could the data be analyzed to compare restrictor-type AN to binge-purge ED? It may be that the polymorphism contributes to some shared behavior such as emotional or impulse dysregulation common to both SUD and bulimia. Alternatively, is there data to support the possibility that this polymorphism contribute to some behavior shared by restrictor AN and BN-AN? It should be noted that some AN also restrict and purge. Does this subgroup express this polymorphism and if so, are they more similar to restricting or binge-purge AN?

As above explained, unfortunately the extended clinical sample does not include any p.T8I positive AN-R or AN-BP patients.

The role of habit in anorexia nervosa remains uncertain as those with AN do not show a generalized deficit in the balance between goal-directed and habitual control of behavior. (see Godier Psych res 244:214-22, 2016).

We agree with Reviewer #2 that the involvement of habits in AN is uncertain. Indeed, it was not uncovered in the paper by Godier et al. However, enhanced habit formation was found in our previous report in a subpopulation of AN-restrictive patient but not in the subgroup of ED-BP patients (see Favier et al., 2020 Journal of Clinical investigation PMID: 33164988). Interestingly, in this article we also described a strong positive correlation between habit formation and cognitive flexibility in AN-R but not in ED-BP. These findings suggest that additional investigation should be conducted to clearly identify which subtype of ED patients have a lack of cognitive flexibility combined with excessive habits.

Alternatively, the role of this polymorphism in anxiety should be considered, as measures of anxiety are elevated in AN independent of state of the illness, and measures of anxiety have been shown to be associated with dorsal striatum dopamine function in AN (Frank Biol Psychiatry PMID 15992780 2005) and anxiety is genetically transmitted in AN and occurs in childhood before the onset of AN symptoms (Meier Eur Eat Disord Rev 2015, PMID: 26347124).

Since we were aware of this point, anxiety behaviors of VGLUT3^{T8I/T8I} mice were thoroughly characterized (see Fig. 4b-e and Fig. S3a-e). We were unable to detect any modification of anxiety in our mouse model. These findings suggest that mood disorder may not be a primary driver of eating disorders but rather a consequence. However, additional investigations are clearly needed to clarify this point.

In summary, these findings could be very important in providing insight into physiological mechanisms in ED but it would be useful to know whether restrictor-type and bulimic-type AN both have this alteration or whether it is specific for bulimic type behaviors (including those with bulimia nervosa) but not anorexia nervosa, given the relationship of bulimia and SUD. The title should be updated to reflect new findings as it seems premature to describe this as a model for AN.

We fully agree with Reviewer #2 that it will be important to determine whether the model and the treatment proposed herein concern restrictor-type and/or bulimic-type AN as well as the relationship with SUD. We hope that this will be the subject of future investigations. We

currently have indirect evidence that the proposed model could be valid beyond p.T8I carriers and in several types of ED. Indeed, in an ongoing investigation we observed that donepezil was able to improve 5 AN-R, 2 AN-BP, 1 BED and 1 BN patients (Case series manuscript in preparation).

We have modified the title as recommended by Reviewer #2

Reviewer #3 (Remarks to the Author):

The manuscript by Favier et al reports on a comprehensive study pinpointing the subtle but important roles played by VGLuT3 expressed by cholinergic interneurons in striatum. Their approach was to study the naturally occurring population of humans with intronic SNP in VGLuT3 at p.T8I, a SNP mutation of the gene in mice (VGLUT3-pT8I) and rescue of (and failure to rescue) VGLuT3-null mice with the VGLuT3-pT8I gene. Their methods spanned from electrophysiology in cultured neurons to a wide battery of behavioral tests , in vitro neurotransmitter release/uptake tests done to synaptic vesicles collected from VGLuT3 null mice, in vivo analyses of VGLUT3 null mice rescued with VGLuT3-pT8I (threonine in position 8, not present in VGLuT1 or 2, mutated to isoleucine), fiber photometry and in vivo voltammetry in striatum to measure ACh release and dopamine (DA) release of WT and VGLuT3T8I/T8I mice, 3D structural analysis of VGLuT3, and a variety of immunocytochemical analyses to assess potential alteration in expression caused by T8I mutation. They report on many data that indicate unperturbed molecular, synaptic and behavioral phenotypes and a handful of findings that could only be appreciated by deeper analyses. The phenotypes that they discovered to be perturbed by VGLuT3 mutations are as follows:

- 1) vesicular synergy (glutamate-dependent ACh uptake Fig 3i; ACh release in Fig 3m-n) are blunted in VGLuT3T8I/T8I mice (N= 7).
- 2) DA transmission is uneven (reduced DA release in DMS but not in DLS) in VGLuT3T8I/T8I mice, (N= 14, Fig 3)
- 3) Enhanced cue-induced vulnerability to cocaine (N=5, Fig 4f-j)
- 4) enhanced habit formation in sucrose seeking (N=12, Fig. 5b) and binge-like drinking of high-sucrose solution during the 1st hour (Fig. 5m, N=10)
- 5) self-starvation that was more severe for the VGLuT3T8I/T8I mice(N=10) and reversal with donepezil (N=8)
- 6) p.T8I variant of VGLuT3 (1.3%, heterozygous, N=8, N=3 for cocaine-addicted patients in Fig 1c) that is associated with higher incidence of psychotic symptoms and SUD (substance use disorder).

These are each very interesting, important, impactful and well-executed as a combination of findings, providing insights into the molecular mechanisms underlying maladaptive behaviors associated with SUD and anorexia nervosa, with implications for pharmacological treatments of both illnesses. The only shortcoming to the entire study is the small sample size of the humans with SNP in threonine 8 (N=3) and also of the animals pointing to the enhanced cue-induced vulnerability to cocaine (N=5).

We are very grateful to Reviewer #3 for her/his positive comments about our work. One important detail is that we found a total of 7 SUDs (not 3) and 1 EDs patients in our extended sample. Of the 7 SUDs patients only 3 were characterized with SAPS scores. This explains the discrepancies between Figure 1c and the text.

Concerning the number of mice that underwent the cocaine self-administration paradigm, we took into consideration comments from Reviewers #1 and #3 and for this new version of the manuscript we ran an additional cohort of mice. We are now presenting solid data with n= 14 WT mice and n= 11 VGLUT3^{T8I/T8I} mice.

They also comment on ONE PATIENT with co-morbidity of eating disorders and drug abuse to suggest that “p.T8I may be associated with both SUDs and EDs”. However, since statistical analyses indicate significances and all parts combine to tell a cohesive story pointing to enhanced habit formation involving VGLUT3 expressed by cholinergic interneurons in dorsomedial striatum, I believe all data should be accepted as presentable.

Once again, we appreciate this supportive vision of our work.

Minor points:

With regard to finding #2 in the list above, the word ‘Uneven,’ which appears in the Abstract was unclear. What I spelled out in parenthesis is clearer (is this what authors meant?), although more wordy.

The abstract was modified as recommended by Reviewer #3 (page 3 lines 66-67).

Page 8. Authors report that VGLUT3 activity accounts for 43% of total striatal vesicular uptake of glutamate. Does this not seem very high, considering the density of cholinergic axons versus glutamatergic axons in striatum? Please comment and relate the new findings to precedents, if there are any.

Indeed, we were also surprised by this finding. But, based on our unpublished anatomical and vesicular uptake observations, we tried to estimate the number of each type of terminals

Figure for Reviewers. Schematic representation of different types of nerve endings and varicosities present in the striatum.

present in the striatum (as illustrated in the figure for Reviewers). The striatum contains 15% of DA (VMAT2-positive) varicosities + 15% of cholinergic (VChT and VGLUT3-positive) varicosities \approx 30% of cortical (VGLUT1-positive) or thalamic (VGLUT2-positive) glutamatergic terminals \approx 40% of GABAergic (VIAAT-positive) terminals.

This approximative estimation suggest that VGLUT3-positive striatal varicosities accounts for \approx half of glutamatergic terminals (15 over 30). Therefore it is not surprising to find a 43% decrease of [3 H]glutamate vesicular uptake in the striatum of VGLUT3-KO.

Some of these estimations are now discussed in Results (page 9 lines 236-241).

Page 9. Why was KCl-depolarization used to evoke DA release? Wouldn't nicotine or glutamate have been better ways to probe for "ACh-glutamate co-transmission ... in controlling DA release"? Please provide a brief rationale for the method used to evoke DA release.

A receptor piloted DA release would necessitate infusion of nicotinic agonist or mGluR antagonist (see Sakae et al. Mol Psychiatry 2015 PMID: 26239290). We estimated that a massive local depolarization by a K⁺ puff could act faster and necessitate a smaller injection volume (200–300 nl) than the infusion of pharmacological compounds to achieve DA release. We previously validated the KCl-depolarization method in 2 publications (Sakae et al. Mol Psychiatry 2015 PMID: 26239290 and Favier et al., PMID: 33164988).

These points are now specified in Material and Methods (page 24, lines 783-786).

The cited reference #66 describing the method used to assess habit formation in obtaining sucrose pellets under devalued versus valued condition was difficult to find. Please add information about the volume and issue to this reference.

Former ref 66 is now ref #73. This problem is due to a conflict between endnote and the Nature Communication format. We manually added the PMID (#22752897) to help reader access this article.

REVIEWER COMMENTS

Reviewer #3 (Remarks to the Author):

My comments to the original version of the MS were relatively minor, compared to other reviewers'. The author attended to my comments adequately.

Reviewer #4 (Remarks to the Author):

In this compelling revised manuscript, Favier et. al. demonstrate a potential neural mechanism for the coordination of maladaptive drug seeking and anorexia-like behaviors across species. The authors have addressed almost all of the previous reviewers comments, with one strong limitation: it is unclear why the authors used male mice only, when the human samples used to elucidate the VGLUT3-pT8I mutation included both males and females. Moreover, despite thoroughly reading through the entire manuscript and supplemental materials, I cannot deduce what percentage of the (relatively small) number of human patients with this mutation are male vs female.

To alleviate these concerns, the authors need to address the following:

1. be clear in the title and abstract that they are studying males only (if this is the case)
2. include more details about the sex and gender (human patients only) of their populations throughout the manuscript (including in figures and results)
3. Why is the SABV region blank in their reporting summary? This needs to be filled out and explained in detail (and honestly gives more reason for concern about this main concern).

Lastly, I tend to agree that the "anorexia-like behaviors" is a bit over stated in the title and throughout this manuscript. The authors mention the one patient who demonstrated both bulimia (which is different from anorexia) and SUD. I think a simple "eating disorders" framework could be used in the title, but to illustrate it as a treatment for all eating disorders, they would also need to perform rescue experiments in their binge eating paradigm (similar to what they did for ABA).

Overall, I recommend revisions to this manuscript, but am enthusiastic about the potential high impact of these findings.

Point-by-point answers to Reviewer's comments.

We would like to express our gratitude to both Reviewers for their positive comments and constructive suggestions. To prepare this new version of the manuscript, we considered all their comments.

Our responses to queries of the Reviewers as well as modifications made to the text are in blue.

General comment

To increase the accuracy of the genetic analysis of human p.T8I (rs45610843) carriers we replaced previous comparison performed with the 1000 genome population by comparison with the gnomAD reference panel v4.0.0, containing 806,886 individuals genotyped by whole-genome array or sequencing (<https://gnomad.broadinstitute.org>) (Supplemental Material page 16-17, lines 199-207). The difference in allelic frequency is now significant (Results, page 6, lines 135-137). However, the SUD sample and the gnomAD reference panel show significant differences in ancestry reflecting our naturalistic recruitment.

Answers to Reviewer #3 remarks to Authors

My comments to the original version of the MS were relatively minor, compared to other reviewers'. The author attended to my comments adequately.

We are very grateful to Reviewer #3 for her/his positive appreciation of the present work.

Answers to Reviewer #4 remarks to Authors

In this compelling revised manuscript, Favier et. al. demonstrate a potential neural mechanism for the coordination of maladaptive drug seeking and anorexia-like behaviors across species. The authors have addressed almost all of the previous reviewers comments, with one strong limitation: it is unclear why the authors used male mice only, when the human samples used to elucidate the VGLUT3-pT8I mutation included both males and females. Moreover, despite thoroughly reading through the entire manuscript and supplemental materials, I cannot deduce what percentage of the (relatively small) number of human patients with this mutation are male vs female.

We apologize for the lack of clarity of the manuscript over this important point. Human demographic data from the SUD cohort are in Supplementary Material (supplementary table S7, p22). The ED cohort included 71 healthy women controls and 270 women out- and inpatients (page 29, lines 948-949). In this modified manuscript we specify in Results the gender of p.T8I carriers (page 6, lines 133-140). There was 2 women and 6 men among the 8 p.T8I carriers.

For this first characterization of the VGLUT3^{T8I/T8I} mice we focused on male mice since we are currently performing an in-depth comparison between male and female VGLUT3^{T8I/T8I} mice. In these

ongoing investigations we found that female and male VGLUT3^{T8I/T8I} mice develop similar compulsive-like grooming in the splash test. However, we also observed that after a chronic social defeat test VGLUT3^{T8I/T8I} male mice have a higher susceptibility to stress than females. We plan to soon perform sucrose binge eating and ABA tests with both sexes that will be published in the near future. Furthermore, we previously reported that hypocholinergic phenotypes are similar in VACHTcKO male and female mice. This is mentioned in Material and Methods (page 20, lines 623-625).

To alleviate these concerns, the authors need to address the following:

1. be clear in the title and abstract that they are studying males only (if this is the case)

The title and the abstract have been modified as requested.

2. include more details about the sex and gender (human patients only) of their populations throughout the manuscript (including in figures and results)

More details about sex and gender of the SUD cohort have been added in Figure 1 (Women are presented with circles and men with squares) and in Results (page 6, lines 133-140) as recommended.

3. Why is the SABV region blank in their reporting summary? This needs to be filled out and explained in detail (and honestly gives more reason for concern about this main concern).

This was a mistake that has been corrected in the present version of the reporting summary.

Lastly, I tend to agree that the "anorexia-like behaviors" is a bit over stated in the title and throughout this manuscript. The authors mention the one patient who demonstrated both bulimia (which is different from anorexia) and SUD. I think a simple "eating disorders" framework could be used in the title, but to illustrate it as a treatment for all eating disorders, they would also need to perform rescue experiments in their binge eating paradigm (similar to what they did for ABA).

We modified the title in keeping with Reviewer #4 recommendation and broadened the relevant discussion elements page 16.

Overall, I recommend revisions to this manuscript, but am enthusiastic about the potential high impact of these findings.

The authors of this work express their gratitude for these positive and supportive recommendation.

REVIEWERS' COMMENTS

Reviewer #4 (Remarks to the Author):

The authors have alleviated my concerns.